# A positive feedback loop between Flower and PI(4,5)P₂ at periactive zones controls bulk endocytosis in Drosophila

Tsai-Ning Li[1], Yu-Jung Chen[1], Ting-Yi Lu[1], You-Tung Wang[1], Hsin-Chieh Lin[1], Chi-Kuang Yao[1,2,3]*

[1]Institute of Biological Chemistry, Academia Sinica, Taipei, Taiwan; [2]Neuroscience Program of Academia Sinica, Academia Sinica, Taipei, Taiwan; [3]Institute of Biochemical Sciences, College of Life Science, National Taiwan University, Taipei, Taiwan

**Abstract** Synaptic vesicle (SV) endocytosis is coupled to exocytosis to maintain SV pool size and thus neurotransmitter release. Intense stimulation induces activity-dependent bulk endocytosis (ADBE) to recapture large quantities of SV constituents in large endosomes from which SVs reform. How these consecutive processes are spatiotemporally coordinated remains unknown. Here, we show that Flower Ca²⁺ channel-dependent phosphatidylinositol 4,5-bisphosphate (PI(4,5)P₂) compartmentalization governs control of these processes in *Drosophila*. Strong stimuli trigger PI(4,5)P₂ microdomain formation at periactive zones. Upon exocytosis, Flower translocates from SVs to periactive zones, where it increases PI(4,5)P₂ levels via Ca²⁺ influxes. Remarkably, PI(4,5)P₂ directly enhances Flower channel activity, thereby establishing a positive feedback loop for PI(4,5)P₂ microdomain compartmentalization. PI(4,5)P₂ microdomains drive ADBE and SV reformation from bulk endosomes. PI(4,5)P₂ further retrieves Flower to bulk endosomes, terminating endocytosis. We propose that the interplay between Flower and PI(4,5)P₂ is the crucial spatiotemporal cue that couples exocytosis to ADBE and subsequent SV reformation.

*For correspondence:
ckyao@gate.sinica.edu.tw

**Competing interests:** The authors declare that no competing interests exist.

## Introduction

Proper synaptic vesicle (SV) exocytosis dictates the robustness of brain activity. Coupling SV exocytosis with proper endocytosis is crucial for maintaining a balance of SV proteins at the release site, plasma membrane equilibrium, SV identity, and SV pool size (*Chanaday et al., 2019*; *Haucke et al., 2011*; *Lou, 2018*; *Wu et al., 2014a*). Currently, four modes of SV endocytosis are proposed. These differ in terms of stimulation intensity for their induction, formation, and molecular components (*Haucke et al., 2011*; *Kononenko and Haucke, 2015*; *Wu et al., 2014a*). Under mild neuronal stimulation, the SV partially fuses with the plasma membrane and reforms at the active zone, the so called 'kiss and run' mode. During clathrin-mediated endocytosis (CME), the SV fully collapses into the plasma membrane, followed by reformation of a single SV. Ultrafast endocytosis was also shown to recycle SVs at a sub-second timescale by forming a ~ 80 nm-sized bulk endosome predominantly at the edge of the active zone. SVs subsequently regenerate from this bulk endosome (*Granseth et al., 2006*; *Watanabe et al., 2013a*; *Watanabe et al., 2013b*; *Zhu et al., 2009*). High frequency stimulation and thus exocytosis could easily surpass the capacity of the three above-described endocytic mechanisms. It has therefore been proposed that activity-dependent bulk endocytosis (ADBE) has the necessary recapture capacity upon intense stimulation (*Clayton et al., 2008*; *Soykan et al., 2017*; *Wu and Wu, 2007*). This retrieval mode is elicited at the periactive zone to recapture large quantities of SV constituents via bulk endosome (~100–500 nm) formation, from which SVs subsequently reform. Hence, specific routes of SV recycling may fit the specific demands

of a wide range of neuronal activities at the synapse. However, how stimulation intensity dictates the choice between these different endocytic modes is not well understood.

ADBE has been documented in many different types of neurons in invertebrates and vertebrates (*Clayton et al., 2008*; *Heerssen et al., 2008*; *Heuser and Reese, 1973*; *Holt et al., 2003*; *Kasprowicz et al., 2008*; *Kittelmann et al., 2013*; *Miller and Heuser, 1984*; *Richards et al., 2000*; *Soykan et al., 2017*; *Stevens et al., 2012*; *Vijayakrishnan et al., 2009*; *Wenzel et al., 2012*; *Wu and Wu, 2007*; *Yao et al., 2017*). Although the precise mechanism regulating ADBE is not known, multiple lines of evidence suggest that actin polymerization may serve as the membrane invagination force responsible for generating bulk endosomes (*Gormal et al., 2015*; *Holt et al., 2003*; *Kokotos and Low, 2015*; *Nguyen et al., 2012*; *Richards et al., 2004*; *Soykan et al., 2017*; *Wu et al., 2016*). Furthermore, phosphatidylinositol metabolism has also been implicated in controlling this recycling mode (*Gaffield et al., 2011*; *Holt et al., 2003*; *Richards et al., 2004*; *Vijayakrishnan et al., 2009*). Importantly, loss of Synaptojanin, the major phosphatidylinositol 4,5-bisphosphate ($PI(4,5)P_2$) catalytic enzyme in neurons (*Tsujishita et al., 2001*), leads to reduced SV endocytosis elicited by intense stimulation, presumably by affecting ADBE (*Mani et al., 2007*). $PI(4,5)P_2$ is known to promote actin polymerization by activating a number of actin modulators (*Janmey et al., 2018*). $PI(4,5)P_2$ is clustered to form microdomains upon the demand for diverse cellular functions (*Aoyagi et al., 2005*; *Chen et al., 2015*; *Honigmann et al., 2013*; *Kabeche et al., 2015*; *Mu et al., 2018*; *Picas et al., 2014*; *Riggi et al., 2018*; *van den Bogaart et al., 2011*). It has also been reported that formation of $PI(4,5)P_2$ microdomains precedes actin polymerization during a process reminiscent of ADBE in neurosecretory cells (*Gormal et al., 2015*). Thus, subcellular compartmentalization of $PI(4,5)P_2$ may provide the spatial information dictating where bulk membranes will invaginate. However, the mechanism initiating the formation of the $PI(4,5)P_2$ microdomains is unknown.

Coordinated SV protein and membrane retrieval plays an important role in maintaining the identity of newly formed SVs during SV recycling (*Kaempf and Maritzen, 2017*; *McMahon and Boucrot, 2011*; *Saheki and De Camilli, 2012*; *Traub and Bonifacino, 2013*). It has been well documented that proper sorting of SV proteins to the nascent SV is achieved in CME by the cooperative action of $PI(4,5)P_2$ and adaptor protein complexes (*Saheki and De Camilli, 2012*; *Traub and Bonifacino, 2013*). Recent studies have also revealed a distinct sorting mechanism that retrieves selective SV cargoes to the bulk endosome via ADBE (*Kokotos et al., 2018*; *Nicholson-Fish et al., 2015*). Several lines of evidence further suggest that clathrin and adaptor protein complexes are required for reforming SVs from the bulk endosome (*Cheung and Cousin, 2012*; *Glyvuk et al., 2010*; *Kokotos et al., 2018*; *Kononenko et al., 2014*; *Park et al., 2016*). A dynaminI/dynaminIII/clathrin-independent mechanism has also been reported as being involved in this process (*Wu et al., 2014c*). Thus, during SV regeneration via ADBE, multiple protein sorting steps may be required to ensure that the SVs harbor the proper compositions of lipids and proteins, thereby endowing specific release probabilities in relation to other modes of endocytosis (*Cheung et al., 2010*; *Hoopmann et al., 2010*; *Nicholson-Fish et al., 2015*; *Silm et al., 2019*). The mechanism by which protein sorting and membrane retrieval are coordinated in this process remains to be explored.

Here, we show that, upon intense stimulation, $PI(4,5)P_2$ is compartmentalized into microdomains at periactive zones in the synaptic boutons of *Drosophila* larval neuromuscular junctions (NMJs). Blockade of $PI(4,5)P_2$ microdomain formation diminishes ADBE and SV reformation from the bulk endosome. Increased intracellular $Ca^{2+}$ and SV exocytosis are prerequisites for initiating ADBE (*Morton et al., 2015*; *Wu and Wu, 2007*). We have previously shown that Flower (Fwe), a SV-associated $Ca^{2+}$ channel, regulates both CME and ADBE, and that its channel activity is strongly activated upon intense stimulation to elicit ADBE (*Yao et al., 2017*). We show that Fwe initiates a positive feedback loop upon $PI(4,5)P_2$ increase to ensure the formation of $PI(4,5)P_2$ microdomains and thus trigger ADBE and subsequent SV reformation. Intriguingly, $PI(4,5)P_2$ also participates in retrieval of Fwe to the bulk endosome, thereby stopping membrane recycling. Hence, spatiotemporal interplays between Flower and $PI(4,5)P_2$ coordinate the retrieval of SV cargos and membranes, coupling exocytosis to ADBE and subsequent SV reformation.

## Results

### Intense neuronal activity induces formation of PI(4,5)P$_2$ microdomains at the presynaptic periactive zone of *Drosophila* synapses

To investigate the dynamics of PI(4,5)P$_2$ in the presynaptic compartment, we expressed a GFP fusion protein of the pleckstrin homology (PH) domain of PLC$_{\delta1}$ (PLC$_{\delta1}$-PH-EGFP) in synaptic boutons of *Drosophila* larval NMJs using *nSyb-GAL4*, a pan-neuronal driver. PLC$_{\delta1}$-PH-EGFP binds to PI(4,5)P$_2$ with high affinity and is widely used to label subcellular compartments in which PI(4,5)P$_2$ is enriched (*Chen et al., 2014*; *Khuong et al., 2010*). We delivered 20 Hz stimuli for three minutes to synaptic boutons in a 2 mM extracellular Ca$^{2+}$ solution and performed live imaging in the third minute. Consecutive snapshot images were taken before stimulation, during the third minute of stimulation, and after stimulation. As shown in *Figure 1a–b*, we observed a very subtle increase in PLC$_{\delta1}$-PH-EGFP fluorescence in individual boutons (white arrows), similar to findings of a previous study (*Verstreken et al., 2009*). However, when we raised the stimulus intensity to 40 Hz, we recorded a robust increase in fluorescence relative to a GFP fusion protein of the plasma membrane-integrated mCD8 domain (*UAS-mCD8-GFP*). Fluorescence signals rapidly returned to basal levels within tens of seconds when the stimuli were removed. We have previously documented that treatment with 40 Hz electric pulses or 90 mM high KCl solution can cause comparable stimulation intensities in *Drosophila* NMJ boutons (*Yao et al., 2017*). High K$^+$ treatment also increased the fluorescence signal of PLC$_{\delta1}$-PH-EGFP. No increase in the presynaptic protein level of PLC$_{\delta1}$-PH-EGFP was found under this condition (*Figure 1—figure supplement 1a–b*), arguing that this stimulation does not induce protein synthesis. These results suggest that, in response to intense stimulation, PLC$_{\delta1}$-PH-EGFP is redistributed and concentrated to PI(4,5)P$_2$-enriched subdomains of the plasma membrane, thereby enhancing the overall fluorescence.

To characterize the subcellular distribution of the stimulus-dependent PI(4,5)P$_2$ induction, we conducted a chemical fixation protocol whereby the NMJ boutons were fixed at rest or immediately after high K$^+$ stimulation and then immunostained for PLC$_{\delta1}$-PH-EGFP using an $\alpha$-GFP antibody to enhance the signal for high-resolution microscopic imaging. Confocal images revealed that, for neurons at rest, native PLC$_{\delta1}$-PH-EGFP fluorescence was weakly detected on the presynaptic plasma membrane labeled by $\alpha$-Hrp staining, with some additional fluorescent signal being dispersed in the cytosol (*Figure 1—figure supplement 1c*). We then elevated PI(4,5)P$_2$ on the plasma membrane by removing a copy of *synaptojanin* (*synj*), which encodes the major neuronal PI(4,5)P$_2$ phosphatase (*Tsujishita et al., 2001*; *Verstreken et al., 2003*). Consistent with previous studies (*Chen et al., 2014*; *Verstreken et al., 2009*), this reduction in Synj levels enhanced PLC$_{\delta1}$-PH-EGFP fluorescence (*Figure 1—figure supplement 1c–d*). In this context, we did not observe a significant change in protein expression of PLC$_{\delta1}$-PH-EGFP in the presynaptic compartment (*Figure 1—figure supplement 1e–f*). By using $\alpha$-GFP immunostaining, these PLC$_{\delta1}$-PH-EGFP signals could be faithfully amplified (*Figure 1—figure supplement 1c–d*). Therefore, this immunostaining approach was subsequently used to monitor presynaptic PI(4,5)P$_2$ levels.

Next, we stimulated the boutons expressing PLC$_{\delta1}$-PH-EGFP with 90 mM K$^+$ and 2 mM Ca$^{2+}$ for 10 min. Similar to our live-imaging results, PLC$_{\delta1}$-PH-EGFP signals were significantly increased on the presynaptic plasma membrane (*Figure 1—figure supplement 1g–h*). In particular, we observed high-level induction of PLC$_{\delta1}$-PH-EGFP puncta (*Figure 1c–d*). To assess the possibility that chemical fixation may have altered membrane properties and consequently plasma membrane PLC$_{\delta1}$-PH-EGFP clustering upon intense stimulation, we further examined the distributions of mCD8-GFP and PLC$_{\delta1}$-PHS39R-EGFP, a PI(4,5)P$_2$-binding mutant (*Khuong et al., 2010*; *Verstreken et al., 2009*). There was no obvious change in the plasma membrane pattern of mCD8-GFP in fixed boutons stimulated with high K$^+$ (*Figure 1—figure supplement 2a–b*). Unlike PLC$_{\delta1}$-PH-EGFP, PLC$_{\delta1}$-PHS39R-EGFP was found mainly in the cytosol at rest. After intense stimulation, immunostaining signals of PLC$_{\delta1}$-PHS39R-EGFP did not accumulate on the plasma membrane (*Figure 1—figure supplement 2c–d*) and were not increased within the boutons (*Figure 1—figure supplement 2e*). Thus, clustering of PLCδ1-PH-EGFP on the plasma membrane upon stimulation results from an increase in local PI(4,5)P$_2$ concentration, although a potential effect of chemical fixation, if any, on PLC$_{\delta1}$-PH-EGFP clustering cannot be excluded. To assess potential dominant-negative effects of PI(4,5)P$_2$ binding by PLC$_{\delta1}$-PH-EGFP on the stimulation-induced accumulation of PI(4,5)P$_2$, we further examined the recruitment of the AP-2 complex by PI(4,5)P$_2$. Similar to PLC$_{\delta1}$-PH-EGFP, levels of the $\alpha$ subunit of

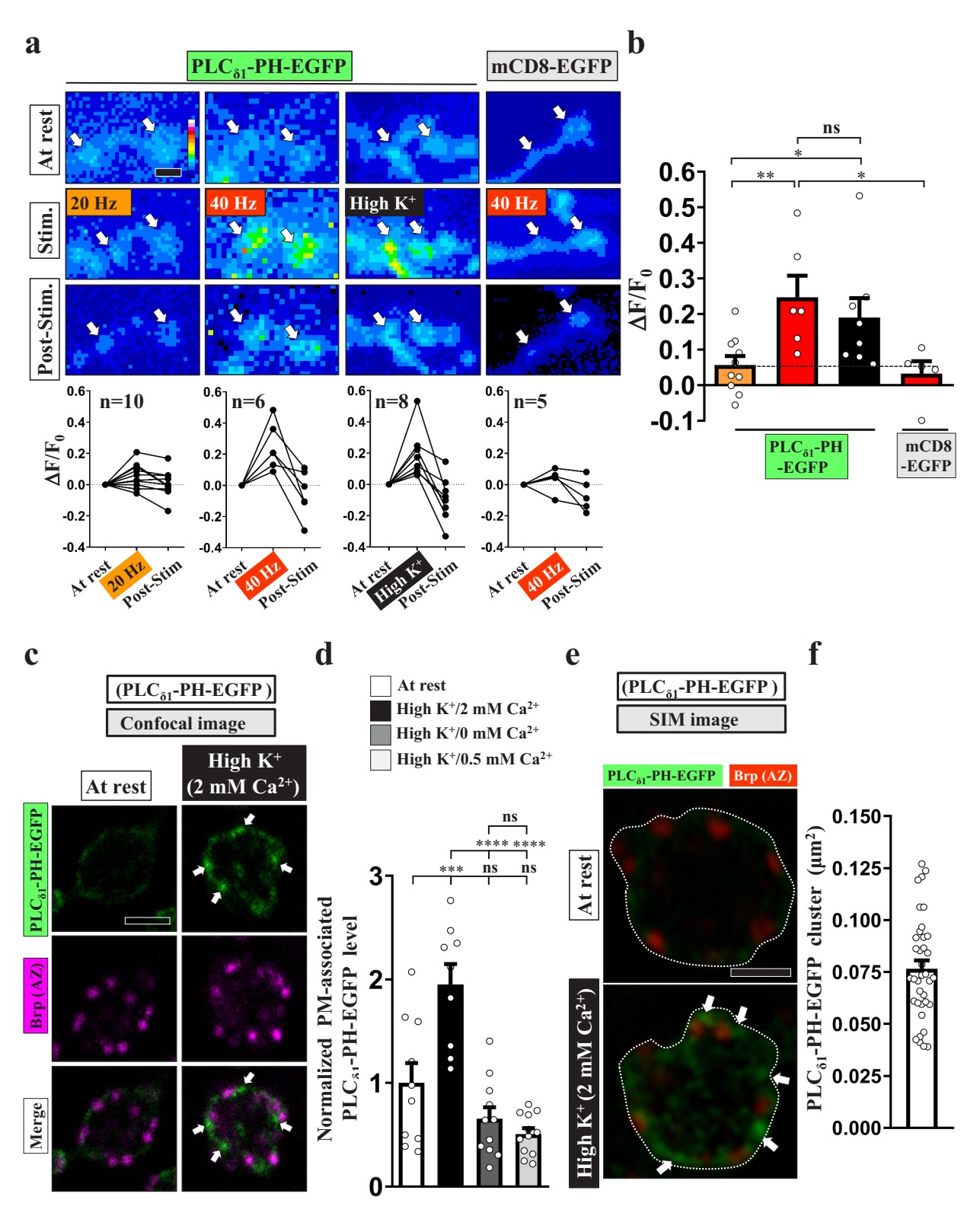

**Figure 1.** PI(4,5)P$_2$ forms microdomains at periactive zones under conditions of intense stimulation. (a–b) Increased fluorescence of PLC$_{\delta 1}$-PH-EGFP but not mCD8-GFP in NMJ boutons upon intense stimulation. (a) (Top) Live images of the boutons (arrows) expressing *UAS-PLC$_{\delta 1}$-PH-EGFP* or *UAS-mCD8-GFP*. The larvae were reared at 25°C. Electrical (20 or 40 Hz) or chemical (90 mM K$^+$) stimulation was conducted in a 2 mM-Ca$^{2+}$ solution for 3 min (electrical) or 5 min (chemical) and then rested in 0 mM Ca$^{2+}$ and 5 mM K$^+$. Snapshot images taken before stimulation, at the third (electrical) and fifth

*Figure 1 continued on next page*

*Figure 1 continued*

(chemical) min of stimulation, and after stimulation. (Bottom) Traces of probe fluorescence for single boutons. The number of boutons imaged (N). (**b**) Quantification data for EGFP fluorescence change. The resting fluorescence level ($F_0$). Fluorescence change evoked by stimulation ($\Delta F$). (**c–f**) $PLC_{\delta 1}$-PH-EGFP enrichment at periactive zones is dependent on $Ca^{2+}$ upon intense stimulation. Single-plane confocal (**c**) or SIM (**e**) images of the boutons expressing $PLC_{\delta 1}$-PH-EGFP. The larvae were reared at 25°C. The boutons subjected to high $K^+$/2 mM $Ca^{2+}$ (10 min stimulation of 90 mM $K^+$/2 mM $Ca^{2+}$), high $K^+$/0 mM $Ca^{2+}$ (10 min stimulation of 90 mM $K^+$/0 mM $Ca^{2+}$), or high $K^+$/0.5 mM $Ca^{2+}$ (1 min stimulation of 90 mM $K^+$/0.5 mM $Ca^{2+}$) treatments were fixed immediately and immunostained for $PLC_{\delta 1}$-PH-EGFP (green) and Bruchpilot (Brp) [an active zone scaffold protein; magenta (**c**), red (**e**)]. The $PLC_{\delta 1}$-PH-EGFP-enriched puncta (Arrows). (**d**) Quantification data for $PLC_{\delta 1}$-PH-EGFP staining intensities, normalized to the value of the resting condition. (**f**) Average area of $PLC_{\delta 1}$-PH-EGFP clusters in individual boutons, as measured by SIM. Individual data values are shown in graphs. p values: ns, not significant; *$p<0.05$; **$p<0.01$; ***$p<0.001$; ****$p<0.0001$. Mean ± SEM. Scale bar: 1 μm (**e**), 2 μm (**a, c**). Statistics: one-way ANOVA with Tukey's post hoc test (**b, d**).

The online version of this article includes the following source data and figure supplement(s) for figure 1:

**Source data 1.** Source data for *Figure 1*.
**Figure supplement 1.** The change in $PLC_{\delta 1}$-PH-EGFP fluorescence is responsible for an increase in the $PI(4,5)P_2$ level.
**Figure supplement 1—source data 1.** Source data for *Figure 1—figure supplement 1*.
**Figure supplement 2.** The distribution of mCD8-EGFP and $PLC_{\delta 1}$-PHS39R-EGFP remains unchanged upon intense stimulation.
**Figure supplement 2—source data 1.** Source data for *Figure 1—figure supplement 2*.
**Figure supplement 3.** $PI(4,5)P_2$ is increased at the periactive zone upon intense stimulation.
**Figure supplement 3—source data 1.** Source data for *Figure 1—figure supplement 3*.

the AP-2 complex (AP-2α) were also increased on the plasma membrane upon high $K^+$ stimulation (*Figure 1—figure supplement 3a–b*). Together, these data indicate that intense stimulation can promote the formation of $PI(4,5)P_2$ microdomains.

We noted that the induced $PI(4,5)P_2$ microdomains were primarily sited at periactive zones marked by activity-dependent localization of Eps15 (*Figure 1c*; *Figure 1—figure supplement 3c*; *Koh et al., 2007*; *Winther et al., 2013*). Periactive zones are known hot-spots for ADBE (*Chanaday et al., 2019*; *Kononenko and Haucke, 2015*; *Wu et al., 2014a*). By using structured illumination microscopy (SIM), we estimated that the average size of these $PI(4,5)P_2$ microdomains formed after stimulation is ~300 nm in diameter (*Figure 1e–f*) (0.07655 ± 0.004 μm$^2$, mean ± S.E.M., n = 37 boutons).

Next, to determine the $Ca^{2+}$ dependence of the $PI(4,5)P_2$ microdomains, we stimulated the boutons in a solution of 90 mM $K^+$ and 0 mM $Ca^{2+}$, which resulted in failure to induce $PI(4,5)P_2$ microdomain formation (*Figure 1d*). We obtained a similar result using 1 min stimulation of 90 mM $K^+$ and 0.5 mM $Ca^{2+}$ (*Figure 1d*), which was previously shown to primarily elicit CME but not ADBE (*Yao et al., 2017*). Hence, these results suggest that intense stimulation can elicit $Ca^{2+}$-driven compartmentalization of $PI(4,5)P_2$ at the periactive zone in synaptic boutons of *Drosophila* NMJs.

## $PI(4,5)P_2$ microdomains are involved in ADBE initiation and SV reformation from bulk endosomes

Next, we investigated the function of $PI(4,5)P_2$ microdomains in ADBE. High $K^+$ treatment is widely used to trigger ADBE in a broad range of synapses (*Akbergenova and Bykhovskaia, 2009*; *Clayton et al., 2008*; *Jin et al., 2019*; *Stevens et al., 2012*; *Vijayakrishnan et al., 2009*; *Wu and Wu, 2007*; *Wu et al., 2014c*). To measure ADBE, we induced ADBE with 90 mM $K^+$ and 2 mM $Ca^{2+}$ for 10 min and then conducted transmission electron microscopy (TEM). ADBE was evoked in wild-type control boutons under these conditions and generated the formation of bulk endosomes (red asterisks, defined as >80 nm in diameter) (*Figure 2c–d*). To suppress the function of $PI(4,5)P_2$, we expressed $PLC_{\delta 1}$-PH-EGFP, anticipating that binding of the $PLC_{\delta 1}$-PH domain to $PI(4,5)P_2$ would restrict availability of $PI(4,5)P_2$ to its effectors and metabolic enzymes (*Figure 2a*; *Khuong et al., 2013*). By using the *GAL4/UAS* system, we were able to adjust expression levels of the $PLC_{\delta 1}$-PH domain via temperature manipulation (*Brand and Perrimon, 1993*; *D'Avino and Thummel, 1999*; *Wilder, 2000*). When we neuronally expressed $PLC_{\delta 1}$-PH-EGFP using *nSyb-GAL4* and grew larvae at 25°C, we found that mild expression of $PLC_{\delta 1}$-PH-EGFP had a mild inhibitory effect on ADBE induction relative to wild-type control boutons (*Figure 2c–d*). In contrast, when larvae were grown at 29°C to effect a two-fold increase in $PLC_{\delta 1}$-PH-EGFP expression (*Figure 2—figure supplement 1a–b*),

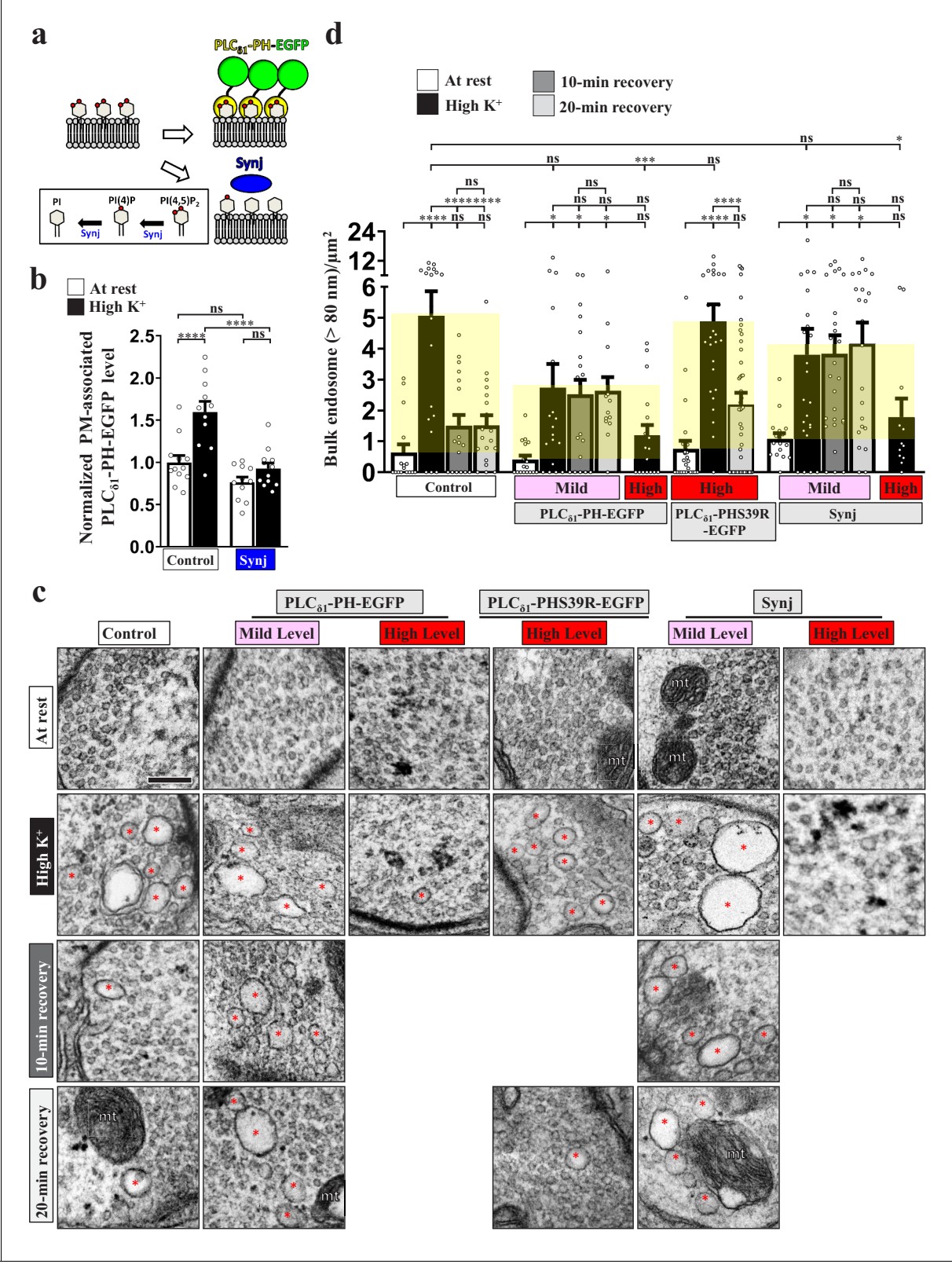

**Figure 2.** PI(4,5)P$_2$ microdomains drive ADBE and SV reformation from bulk endosomes. Reducing PI(4,5)P$_2$ availability suppresses ADBE and subsequent SV reformation. (a) A schematic for PI(4,5)P$_2$ suppression by PLC$_{\delta 1}$-PH-EGFP or Synj expression. (b) Expression of Synj reduced presynaptic plasma membrane PI(4,5)P$_2$ upon high K$^+$ treatment. The boutons co-expressing *UAS-PLC$_{\delta 1}$-PH-EGFP* with *UAS-RFP* (control) or *UAS-synj* using *nSyb-GAL4* were reared at 25˚C and subjected to resting condition (10 min incubation of 5 mM K$^+$/0 mM Ca$^{2+}$) or high K$^+$ stimulation (10 min stimulation of

*Figure 2 continued*

90 mM $K^+$/2 mM $Ca^{2+}$), followed by α-GFP immunostaining. Single-plane confocal images of the boutons are shown in *Figure 2—figure supplement 1c*. Quantification data for PLC$_{\delta 1}$-PH-EGFP staining intensity are shown, normalized to the value of the resting condition of controls. (c) TEM images of the boutons of controls (*nSyb-GAL4/+* at 29°C), mild PLC$_{\delta 1}$-PH-EGFP expression (*nSyb-GAL4/UAS-PLC$_{\delta 1}$-PH-EGFP* at 25°C), high PLC$_{\delta 1}$-PH-EGFP expression (*nSyb-GAL4/UAS-PLC$_{\delta 1}$-PH-EGFP* at 29°C), high PLC$_{\delta 1}$-PHS39R-EGFP expression (*nSyb-GAL4/UAS-PLC$_{\delta 1}$-PHS39R-EGFP* at 29°C), mild Synj expression (*nSyb-GAL4/UAS-synj* at 25°C), or high Synj expression (*nSyb-GAL4/UAS-synj* at 29°C). At rest (10 min incubation of 5 mM $K^+$/0 mM $Ca^{2+}$). High $K^+$ (10 min stimulation of 90 mM $K^+$/2 mM $Ca^{2+}$). 10 min recovery (10 min stimulation of 90 mM $K^+$/2 mM $Ca^{2+}$, followed by 10 min incubation of 5 mM $K^+$/0 mM $Ca^{2+}$). 20 min recovery (10 min stimulation of 90 mM $K^+$/2 mM $Ca^{2+}$, followed by 20 min incubation of 5 mM $K^+$/0 mM $Ca^{2+}$). Bulk endosomes (>80 nm in diameter, red asterisks). Mitochondria (mt). Quantification data for total number of bulk endosomes per bouton area (d). Individual data values are shown in graphs. p values: ns, not significant; *p<0.05; **p<0.01; ***p<0.001; ****p<0.0001. Mean ± SEM. Scale bar: 500 nm. Statistics: one-way ANOVA with Tukey's post hoc test.

The online version of this article includes the following source data and figure supplement(s) for figure 2:

**Source data 1.** Source data for *Figure 2*.
**Figure supplement 1.** Blockade of the PI(4,5)P$_2$ microdomains by expressing PLC$_{\delta 1}$-PH-EGFP and Synj.
**Figure supplement 1—source data 1.** Source data for *Figure 2—figure supplement 1*.
**Figure supplement 2.** Blockade of the PI(4,5)P$_2$ microdomains abolishes SV reformation from the bulk endosome.
**Figure supplement 2—source data 1.** Source data for *Figure 2—figure supplement 2*.

ADBE was almost completely abolished (*Figure 2c–d*). However, when larvae expressing PLC$_{\delta 1}$-PHS39R-EGFP were grown at 29°C, ADBE was not suppressed by the mutant protein (*Figure 2c–d*).

Synj comprises a central 5-phosphatase domain that specifically dephosphorylates the 5' position of PI(4,5)P$_2$ to produce PI(4)P (*McPherson et al., 1996*; *Woscholski et al., 1997*). In addition, an N-terminal Sac1 domain converts several phosphatidylinositides—including PI(3,5)P$_2$, PI(3)P, and PI(4)P—to PI (*Figure 2a*; *Guo et al., 1999*). Indeed, overexpression of Synj reduced the formation of the PI(4,5)P$_2$ microdomains induced by high $K^+$ (*Figure 2b*; *Figure 2—figure supplement 1c*). Importantly, overexpression of Synj resulted in dosage-dependent inhibition of ADBE (*Figure 2c–d*), similar to the effect of increasing levels of the PLC$_{\delta 1}$-PH domain. Together, these data suggest that the PI(4,5)P$_2$ microdomains initiate ADBE to control SV membrane retrieval upon intense stimulation.

SVs regenerate from bulk endosomes within minutes of their formation (*Cheung and Cousin, 2012*; *Glyvuk et al., 2010*; *Kononenko et al., 2014*; *Stevens et al., 2012*; *Wu et al., 2014c*). Using an approach employed previously (*Stevens et al., 2012*), we treated the boutons with high $K^+$ followed by an incubation in 5 mM $K^+$ and 0 mM $Ca^{2+}$ solution for 10 or 20 min, allowing the SVs to reform from the bulk endosomes. In controls (*nSyb-GAL4*), the number and area of the induced bulk endosomes reverted to almost basal levels within 10 min (*Figure 2c–d*; *Figure 2—figure supplement 2*), demonstrating proper SV reformation. Interestingly, upon expression of low levels of PLC$_{\delta 1}$-PH-EGFP, the induced bulk endosomes remained after 10 min and 20 min recovery periods, yet ADBE was only partially impaired (*Figure 2c–d*; *Figure 2—figure supplement 2*). In contrast, SV reformation actively occurred in the boutons expressing high levels of PLC$_{\delta 1}$-PHS39R-EGFP (*Figure 2c–d*; *Figure 2—figure supplement 2*). Furthermore, we observed a loss of SV reformation ability upon mild overexpression of Synj (*Figure 2c–d*; *Figure 2—figure supplement 2*). Hence, the formation of PI(4,5)P$_2$ microdomains is essential for both ADBE and SV reformation from the bulk endosome, with the latter event being elicited by relatively high levels of PI(4,5)P$_2$. Note that perturbation of PI(4,5)P$_2$ microdomain formation did not cause SV accumulation on the bulk endosome during recovery periods, indicating that PI(4,5)P$_2$ microdomains may play an early role in SV reformation from bulk endosomes.

## PI(4,5)P$_2$ microdomains are established via a positive feedback loop of fwe and PI(4,5)P$_2$

We have previously shown that the SV-associated $Ca^{2+}$ channel Flower (Fwe) elevates presynaptic $Ca^{2+}$ levels in response to strong stimuli to trigger ADBE (*Yao et al., 2009*; *Yao et al., 2017*). Given the $Ca^{2+}$ dependence of PI(4,5)P$_2$ microdomains (*Figure 1d*), we hypothesized that exocytosis evoked by intense stimulation promotes Fwe clustering at periactive zones, where it may provide the $Ca^{2+}$ influx to induce PI(4,5)P$_2$ microdomain formation. To test this hypothesis, we first conducted a proximity ligation assay (PLA) (*Söderberg et al., 2008*) to investigate if there is a close association between Fwe and PI(4,5)P$_2$ in response to stimulation. In our PLA (*Figure 3a*), *UAS-Flag-*

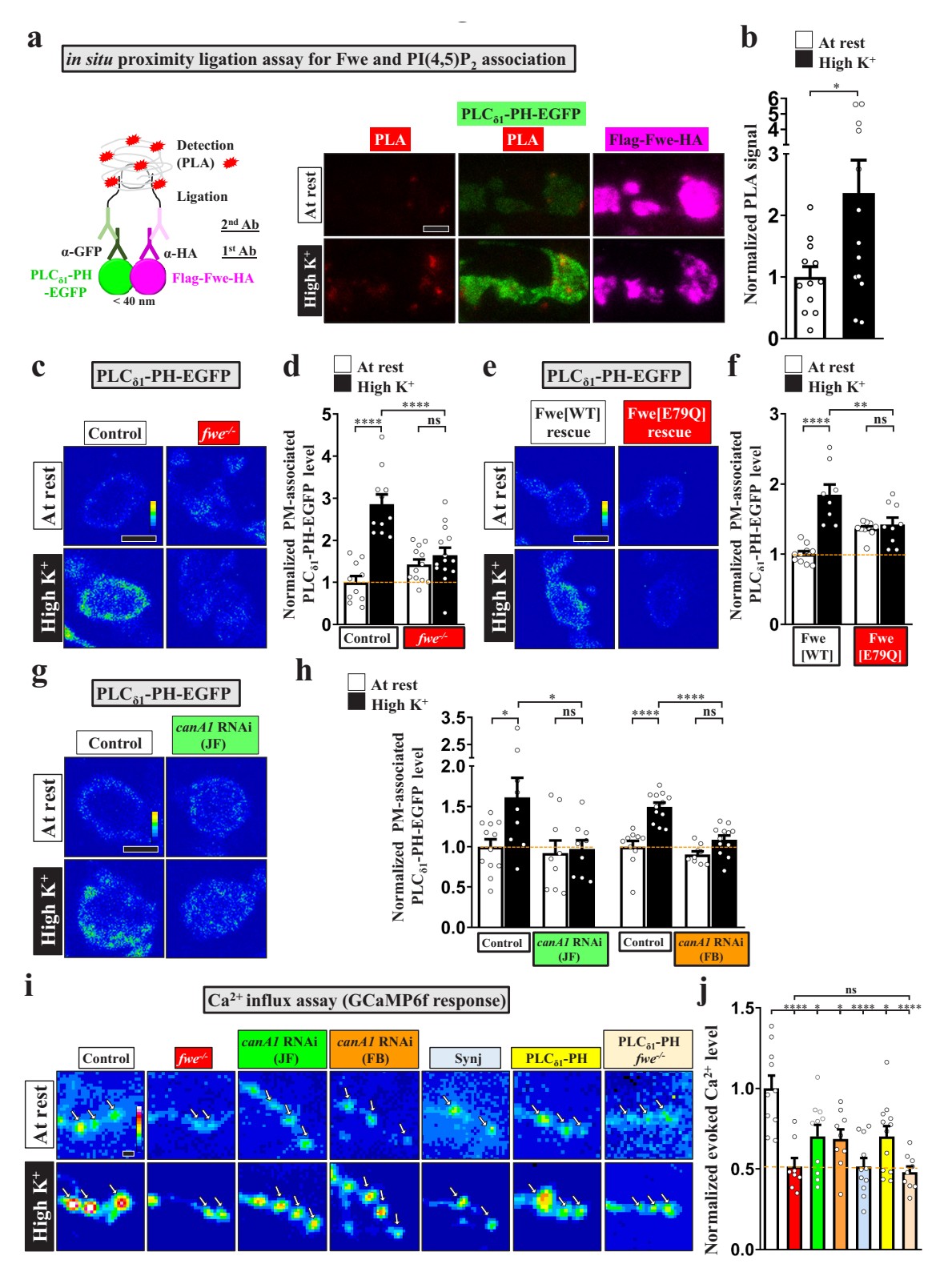

**Figure 3.** Fwe and PI(4,5)P$_2$ form a positive feedback loop to establish PI(4,5)P$_2$ microdomains. (a–b) Fwe and PLC$_{\delta1}$-PH-EGFP interact upon intense stimulation. (a) (Left) A schematic for PLA. (Right) Z-projected confocal images of *fwe* mutant boutons co-expressing Flag-Fwe-HA and PLC$_{\delta1}$-PH-EGFP (*nSyb-GAL4/UAS-Flag-fwe-HA/UAS-PLC$_{\delta1}$-PH-EGFP in fwe$^{DB25/DB56}$* at 25°C). After resting condition or high K$^+$ stimulation, the boutons were subjected to PLA. α-GFP and α-HA stained for PLC$_{\delta1}$-PH-EGFP (green) and Flag-Fwe-HA (magenta), respectively. PLA signals (red). (b) Quantification data for PLA

*Figure 3 continued on next page*

**Figure 3 continued**

signal intensities, normalized to the value of the resting condition. (c–f) Loss of Fwe or its $Ca^{2+}$ channel activity perturbs $PI(4,5)P_2$ microdomain formation. (c and e). (c) Single-plane confocal images of the boutons neuronally expressing $UAS\text{-}PLC_{\delta 1}\text{-}PH\text{-}EGFP$ in $fwe^{DB25/+}$ (control) or $fwe^{DB25/DB56}$ at 25°C. The boutons were subjected to resting condition or high $K^+$ stimulation, followed by α-GFP immunostaining. Quantification data for $PLC_{\delta 1}\text{-}PH\text{-}EGFP$ staining intensity are shown, normalized to the value of the resting condition of controls (d). (e) Single-plane confocal images of the boutons neuronally expressing $lexAop2\text{-}PLC_{\delta 1}\text{-}PH\text{-}EGFP$ using $vglut\text{-}lexA$ in wild-type Fwe rescue boutons ($vglut\text{-}lexA/lexAop2\text{-}PLC_{\delta 1}\text{-}PH\text{-}EGFP$, $nSyb(w)\text{-}GAL4/UAS\text{-}flag\text{-}fwe\text{-}RB\text{-}HA$ in $fwe^{DB25/fweDB56}$ at 25°C), or FweE79Q rescue boutons ($vglut\text{-}lexA/lexAop2\text{-}PLC_{\delta 1}\text{-}PH\text{-}EGFP$, $nSyb(w)\text{-}GAL4/UAS\text{-}flag\text{-}fweE79Q\text{-}RB\text{-}HA$ in $fwe^{DB25/fweDB5}$ at 25°C). When the weak $nSyb(w)\text{-}GAL4$ driver drove low expression of $UAS\text{-}fwe$ transgenes, this binary system ($vglut\text{-}lexA/lexAop2\text{-}PLC_{\delta 1}\text{-}PH\text{-}EGFP$) was used to produce detectable levels of $PLC_{\delta 1}\text{-}PH\text{-}EGFP$ in boutons. Quantification data for $PLC_{\delta 1}\text{-}PH\text{-}EGFP$ staining intensity are shown, normalized to the value of the resting condition of Fwe-rescued boutons (f). (g–h) $canA1$ RNAi knockdown impairs $PI(4,5)P_2$ microdomain formation. Single-plane confocal images of the boutons co-expressing $UAS\text{-}PLC_{\delta 1}\text{-}PH\text{-}EGFP$ with $UAS\text{-}RFP$ (control), $UAS\text{-}canA1\text{-}RNAi$ (TRiP.JF01871), or $UAS\text{-}canA1\text{-}RNAi$ (FB4) using $nSyb\text{-}GAL4$. The larvae were reared at 25°C. After resting condition or high $K^+$ stimulation, the boutons were stained with α-GFP. (h) Quantification data for $PLC_{\delta 1}\text{-}PH\text{-}EGFP$ staining intensities, normalized to the value of the resting condition of controls. (i–j) Blockade of the $PI(4,5)P_2$ microdomains attenuates Fwe $Ca^{2+}$ conductance. (i) Snapshot $Ca^{2+}$ images of the boutons (arrows) expressing $lexAop2\text{-}GCaMP6f$ using $vglut\text{-}lexA$. The larvae of control ($w^{1118}$), fwe mutant ($fwe^{DB25/DB56}$), $canA1$ RNAi (JF) ($nSyb\text{-}GAL4/UAS\text{-}canA1\text{-}RNAi$ (TRiP.JF01871)), $canA1$ RNAi (FB) ($nSyb\text{-}GAL4/UAS\text{-}canA1\text{-}RNAi$ (FB4)), Synj overexpression ($nSyb\text{-}GAL4/UAS\text{-}synj$), $PLC_{\delta 1}\text{-}PH\text{-}APEX2\text{-}HA$ expression ($vglut\text{-}lexA/LexAop2\text{-}PLC_{\delta 1}\text{-}PH\text{-}APEX2\text{-}HA$), or fwe mutant expressing $PLC_{\delta 1}\text{-}PH\text{-}APEX2\text{-}HA$ ($vglut\text{-}lexA/LexAop2\text{-}PLC_{\delta 1}\text{-}PH\text{-}APEX2\text{-}HA$ in $fwe^{DB25/DB56}$) were reared at 25°C. Imaging was taken in the fifth minute for one minute after high $K^+$ (2 mM $Ca^{2+}$) stimulation. (j) Quantification data for evoked $Ca^{2+}$ level, normalized to the value of controls. Evoked $Ca^{2+}$ levels are shown as the increase in GCaMP6f fluorescence under high $K^+$ stimulation. Individual data values are shown in graphs. p values: ns, not significant; *$p<0.05$; **$p<0.01$; ****$p<0.0001$. Mean ± SEM. Scale bar: 2 μm. Statistics: Student $t$-test (b). One-way ANOVA with Tukey's post hoc test (d, f, h, j).

The online version of this article includes the following source data for figure 3:

**Source data 1.** Source data for *Figure 3*.

Fwe-HA was expressed in a *fwe* mutant background to replace endogenous Fwe protein with a tagged protein, and expression of $UAS\text{-}PLC_{\delta 1}\text{-}PH\text{-}EGFP$ reported the localization of $PI(4,5)P_2$. Primary antibodies against the HA tag and EGFP protein were used to detect interactions between Flag-Fwe-HA and $PLC_{\delta 1}\text{-}PH\text{-}EGFP$. The PLA signal was low in resting boutons, whereas high $K^+$ treatment significantly increased PLA signal intensity (*Figure 3a–b*). Therefore, these results suggest that Fwe and $PI(4,5)P_2$ are closely colocalized when intense stimulation triggers $PI(4,5)P_2$ microdomain formation.

Next, we investigated the effect of loss of Fwe on $PI(4,5)P_2$ microdomain formation. Basal levels of $PI(4,5)P_2$ were not affected in the *fwe* mutant relative to control (*Figure 3c–d*). However, intense stimulation failed to elicit $PI(4,5)P_2$ microdomain formation in the *fwe* mutant (*Figure 3c–d*), revealing a crucial role for Fwe in this process. To determine if the $Ca^{2+}$ channel activity of Fwe is responsible for this activity, we conducted rescue experiments based on our previous report in which *fwe* mutant boutons exhibited expression of the wild-type Fwe transgene or the FweE79Q mutant transgene that has reduced $Ca^{2+}$ conductance (*Yao et al., 2017*). We found that the wild-type Fwe transgene promoted $PI(4,5)P_2$ microdomain formation, whereas the FweE79Q mutant transgene lost that ability (*Figure 3e–f*). Thus, Fwe triggers the formation of $PI(4,5)P_2$ microdomains in a $Ca^{2+}$ channel-dependent manner.

Calmodulin and Calcineurin are thought to be the $Ca^{2+}$ sensors for ADBE (*Evans and Cousin, 2007*; *Jin et al., 2019*; *Marks and McMahon, 1998*; *Sun et al., 2010*; *Wu et al., 2009*; *Wu et al., 2014b*). The Calcineurin complex dephosphorylates $PI(4,5)P_2$ metabolic enzymes, including Synj and Phosphatidylinositol 4-phosphate 5-kinase Iγ (PIPKIγ) (*Cousin and Robinson, 2001*; *Lee et al., 2005*; *Lee et al., 2004*; *van den Bout and Divecha, 2009*). *Drosophila* possesses three isoforms of the catalytic subunit of Calcineurin, i.e., CanA1, CanA-14F, and Pp2B-14D. CanA1 regulates development of *Drosophila* NMJ boutons (*Wong et al., 2014*). Therefore, we knocked down *canA1* in neurons by using two independent $UAS\text{-}canA1\text{-}RNAi$ constructs, $UAS\text{-}canA1\text{-}RNAi(JF01871)$ (*Wong et al., 2014*) and $UAS\text{-}canA1\text{-}RNAi(FB4)$ (*Dijkers and O'Farrell, 2007*). Similar to the effect of loss of Fwe, reducing CanA1 levels via expression of either RNAi construct greatly suppressed the formation of $PI(4,5)P_2$ microdomains compared to the control (*Figure 3g–h*), indicating that Calcineurin mediates the $Ca^{2+}$ influx conducted by Fwe to induce $PI(4,5)P_2$ microdomains.

Our previous $Ca^{2+}$ imaging data showed that intense stimulation activates Fwe to increase presynaptic $Ca^{2+}$ concentrations (*Yao et al., 2017*). We measured intracellular $Ca^{2+}$ levels with the $Ca^{2+}$ indicator GCaMP6f (*Chen et al., 2013*), and found that whereas control boutons exhibited a robust

increase in intracellular $Ca^{2+}$ upon high $K^+$ stimulation, *fwe* mutant boutons exhibit an impaired $Ca^{2+}$ response (*Figure 3i–j*). Unexpectedly, *canA1* knockdown also elicited the same deficient $Ca^{2+}$ response (*Figure 3i–j*). Similar suppressive effects were obtained upon overexpressing Synj or the $PLC_{\delta 1}$-PH domain (*Figure 3i–j*). Hence, Calcineurin activation may increase $PI(4,5)P_2$ activity, which may in turn promote the $Ca^{2+}$ channel activity of Fwe to further increase intracellular $Ca^{2+}$ levels. To investigate the potential for such feedback regulation, we expressed the $PLC_{\delta 1}$-PH domain in a loss of Fwe background. Expression of the $PLC_{\delta 1}$-PH domain did not rescue the low $Ca^{2+}$ concentration caused by the *fwe* mutation (*Figure 3i–j*), showing that the $Ca^{2+}$ suppression exerted by the $PLC_{\delta 1}$-PH domain indeed depends on Fwe. These findings support that a positive feedback loop involving Fwe and $PI(4,5)P_2$ is responsible for the formation of $PI(4,5)P_2$ microdomains.

## $PI(4,5)P_2$ gates *fwe*

A well-known function of $PI(4,5)P_2$ is to modulate ion channel activity through its electrostatic binding to clustered positively-charged amino acids adjacent to the transmembrane domains of ion channels (*Hille et al., 2015*; *Suh and Hille, 2008*). Through protein alignment analysis, we found tandem positively-charged amino acids, including lysine (K) and arginine (R), in the intracellular juxta-transmembrane regions of Fwe. These residues are evolutionarily conserved in mice and humans (*Figure 4a*), whereas other cytosolic residues show poor conservation. This feature inspired us to test the potential impact of $PI(4,5)P_2$ on the channel function of Fwe. To determine direct interaction between Fwe and $PI(4,5)P_2$, we conducted a nanoluciferase (Nluc)-based bioluminance resonance energy transfer (BRET) assay (*Cabanos et al., 2017*). In our BRET assay (*Figure 4a*), upon ion channel binding of BODIPY-TMR-conjugated $PI(4,5)P_2$, illumination of Nluc-fused ion channels wrapped in detergent-formed micelles can excite BODIPY-TMR-conjugated $PI(4,5)P_2$ to emit a BRET signal. As shown in *Figure 4b*, we reconstituted the micelles containing purified Nluc-Fwe-1D4 fusion proteins and, after adding BODIPY-TMR-$PI(4,5)P_2$ and furimazine (a Nluc substrate), we observed a remarkable increase in BRET signal. Excess cold $PI(4,5)P_2$ reduced the signal to ~50% of the BRET signals by competing for the $PI(4,5)P_2$ binding sites in Fwe, suggesting direct $PI(4,5)P_2$ binding to Fwe. To assess the involvement of the positively-charged amino acids of Fwe in $PI(4,5)P_2$ binding, we mutated all of the clustered positively-charged amino acids to non-charged alanine to eliminate the electrostatic interactions. Residue substitution resulted in a significant reduction in BRET signal, comparable to the competitive effect attributable to provision of excess cold $PI(4,5)P_2$. Therefore, the majority of the $PI(4,5)P_2$ binding activity of Fwe is mediated by these positively-charged amino acids. Specifically, alanine substitution of residues in both the middle (K95, K100, R105) and C-terminal (K146, K147, R150) regions of Fwe reduced $PI(4,5)P_2$-specific binding (*Figure 4b–c*). Moreover, when N-terminal residues (K29, R33) of Fwe were further substituted with alanines, binding of $PI(4,5)P_2$ was also reduced (*Figure 4b–c*). Hence, these in vitro assays reveal that Fwe directly binds $PI(4,5)P_2$ through multiple regions.

To directly test how $PI(4,5)P_2$ affects Flower $Ca^{2+}$ channel function, we generated *UAS* transgenes for the Fwe variants in which all or subsets of positively-charged amino acids were mutated to alanine and performed a mutant rescue experiment using *nSyb-GAL4*. Mutations of all nine residues or only those in the middle region (K95/K100/R105) led to very low protein expression levels, preventing further study. However, alanine substitution of C-terminal residues K146/K147/R150 did not affect SV localization of Fwe or its ability to regulate presynaptic $Ca^{2+}$ concentration and induce $PI(4,5)P_2$ microdomain formation (*Figure 4—figure supplement 1a–f*), suggesting that these residues do not play a regulatory role in Fwe channel activity. We also mutated the N-terminal residues K29/R33. The resulting K29A/R33A variant was still able to properly localize to presynaptic terminals (*Figure 4—figure supplement 2a–b*). However, upon high $K^+$ stimulation, the K29A/R33A variant lost that ability to maintain proper intracellular $Ca^{2+}$ levels (*Figure 4d–e*). Moreover, that variant failed to promote $PI(4,5)P_2$ microdomain formation upon high $K^+$ stimulation (*Figure 4f–g*). These results reveal that the positive feedback loop involving Fwe and $PI(4,5)P_2$ relies on $PI(4,5)P_2$-dependent gating control of Fwe.

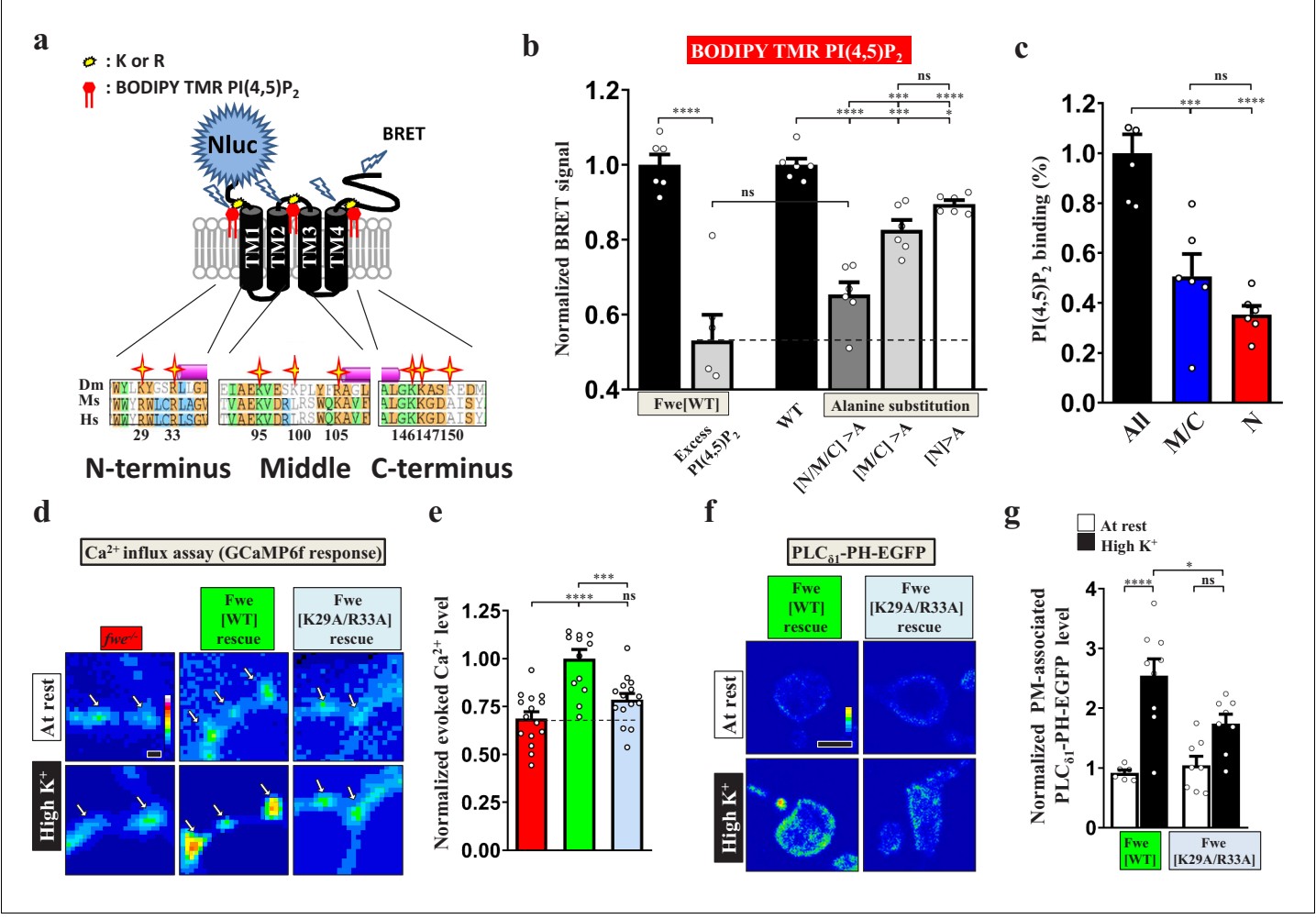

**Figure 4.** PI(4,5)P$_2$ binds to Fwe and promotes its Ca$^{2+}$ channelling activity. (a–c) PI(4,5)P$_2$ binds Fwe. (a) A schematic of the Fwe structure and a BRET assay. Stars highlight conserved lysine (K) and arginine (R) in juxta-transmembrane regions: N-terminus (N), middle-domain (M), and C-terminus (C). *Drosophila melanogaster* (Dm). *Mus Musculus* (Ms). *Human Sapiens* (Hs). N-terminal fusion of Nluc to Fwe allows detection of PI(4,5)P$_2$ binding. A 1D4 epitope was used for protein purification. (b) Nluc-Fwe-1D4 in micelles excites BODIPY-TMR-conjugated PI(4,5)P$_2$ to emit BRET signal, which is decreased by competitive cold PI(4,5)P$_2$ (1 mM). Alanine substitution of positively-charged residues in all regions (N/M/C > A), both middle-domain and C-terminus (M/C > A), or N-terminus (N > A) reduced BRET signals. (c) Corresponding PI(4,5)P$_2$ binding ability was calculated by subtracting the signal values of N/M/C > A, M/C > A or N > A from that for WT. Quantification data was normalized to the signal value of all mutations (N/M/C). (d) PI(4,5)P$_2$ controls Fwe channel activity. Snapshot Ca$^{2+}$ images of the boutons (arrows) expressing *lexAop2-GCaMP6f* using *vglut-lexA. fwe* mutant (*fwe$^{DB25/DB56}$*). HA-Fwe[WT]-APEX2 rescue (*nSyb-GAL4/UAS-HA-Fwe[WT]-APEX2 in fwe$^{DB25/DB56}$*). HA-Fwe[K29A/R33A]-APEX2 rescue (*nSyb-GAL4/UAS-HA-Fwe[K29A/R33A]-APEX2 in fwe$^{DB25/DB56}$*). The larvae were reared at 25°C. Images were taken in the fifth minute for one minute after high K$^+$ stimulation. (e) Quantification data for evoked Ca$^{2+}$ level, normalized to the value of HA-Fwe-APEX2 rescue larvae. (f) Single-plane confocal images of the boutons expressing *UAS-PLC$_{δ1}$-PH-EGFP*. The larvae were reared at 25°C. After resting conditions or high K$^+$ stimulation, the boutons were stained with α-GFP. (g) Quantification data for the PLC$_{δ1}$-PH-EGFP staining intensity, normalized to the value of the resting condition of HA-Fwe-APEX2 rescue larvae. Individual data values are shown in graphs. p values: ns, not significant; *$p < 0.05$; ***$p < 0.001$; ****$p < 0.0001$. Mean ± SEM. Scale bar: 2 μm (d, f). Statistics: Student *t*-test (b). One-way ANOVA with Tukey's post hoc test (b, c, e, g).

The online version of this article includes the following source data and figure supplement(s) for figure 4:

**Source data 1.** Source data for *Figure 4*.
**Figure supplement 1.** The C-terminal residues (K146/K147/R150) of Fwe are not involved in regulating Ca$^{2+}$ channel activity.
**Figure supplement 1—source data 1.** Source data for *Figure 4—figure supplement 1*.
**Figure supplement 2.** Expression of Fwe[R29A/K33A] in boutons.
**Figure supplement 2—source data 1.** Source data for *Figure 4—figure supplement 2*.

## Blockade of the positive feedback loop reduces ADBE and SV reformation from bulk endosomes

Next, we assessed the impact of the Fwe and PI(4,5)P$_2$ regulatory feedback loop on ADBE. Loss of Fwe severely impaired formation of bulk endosomes induced by ADBE under high K$^+$ conditions compared to the *fwe* mutant rescue control (*Figure 5a–b*), consistent with our previous findings (*Yao et al., 2017*). Next, we found that expression of wild-type Fwe protein or the K146A/K147A/R150A mutant variant restored proper ADBE in the *fwe* mutant background (*Figure 5a–b*; *Figure 4—figure supplement 1g–h*), whereas expression of the K29A/R33A variant failed to rescue ADBE deficiency (*Figure 5a–b*). Consistent with the suppressive effects caused by expression of the PLCδ1-PH domain or Synj (*Figure 2*), all of the bulk endosomes that remained in boutons lacking *fwe* could not generate new SVs during a 10 min or even 20 min recovery period (*Figure 5a–b*; *Figure 5—figure supplement 1*). Expression of wild-type Fwe protein but not the K29A/R33A mutant variant rescued this defect (*Figure 5a–b*; *Figure 5—figure supplement 1*).

Neuronal *canA1* knockdown by expressing *canA1* RNAi construct also impaired ADBE relative to the *nSyb-GAL4* control (*Figure 5a–b*). To verify if CanA1 regulates ADBE in an Fwe-dependent manner, we overexpressed Fwe to increase the intracellular Ca$^{2+}$ levels under *canA1 RNAi* knockdown conditions. Fwe overexpression significantly reversed the ADBE defect (*Figure 5a–b*), suggesting that increasing Fwe-dependent Ca$^{2+}$ influx can augment activation of the residual CanA1 enzymes and thus normalize downstream ADBE. Taken together with our results reported in previous sections, we propose that the positive feedback loop involving Fwe, CanA1 and PI(4,5)P$_2$ compartmentalizes PI(4,5)P$_2$ microdomains at the periactive zone of boutons to dictate and coordinate ADBE and subsequent SV reformation.

## PI(4,5)P$_2$ facilitates retrieval of fwe to bulk endosomes

It was reported recently that a SV protein sorting process occurs during ADBE (*Kokotos et al., 2018*; *Nicholson-Fish et al., 2015*). VAMP4, a v-SNARE protein, is essential for ADBE to proceed, and it is selectively retrieved by ADBE (*Kokotos et al., 2018*; *Nicholson-Fish et al., 2015*). Given the important role of Fwe in triggering ADBE, we wondered if Fwe is sorted to bulk endosomes during ADBE. To visualize the vesicular localization of Fwe, we rescued the *fwe* mutant by expressing a *UAS* transgene of the APEX2 fusion protein of Fwe (*UAS-HA-Fwe-APEX2*), and then conducted diaminobenzidine (DAB) labeling and TEM. By means of confocal microscopy, we observed that HA-Fwe-APEX2 immunostaining signals were properly present on the SVs marked by Syt and Csp immunostaining (*Figure 6—figure supplement 1a*). Furthermore, expression of this fusion protein was able to rescue the endocytic defects (*Figures 4* and *5*) and early animal lethality (not shown) caused by loss of *fwe*. Therefore, HA-Fwe-APEX2 is functionally equivalent to endogenous Fwe. APEX2 is an engineered peroxidase that is capable of catalyzing DAB polymerization and proximal deposition, with the DAB polymers binding electron-dense osmium to enhance electron microscopy contrast (*Lam et al., 2015*). Whereas no DAB staining signals were observed in SVs in the Flag-Fwe-HA rescue control boutons stimulated by high K$^+$ (yellow arrows, *Figure 6—figure supplement 1b'*; *Figure 6a*), the SV-based localization of HA-Fwe-APEX2 was clearly revealed by DAB staining in HA-Fwe-APEX2-rescued boutons under the resting and stimulation conditions (yellow arrows, *Figure 6—figure supplement 1c–d*). Moreover, specific DAB labeling also revealed the presynaptic plasma membrane localization of HA-Fwe-APEX2 (white arrows, *Figure 6—figure supplement 1b–c*), an outcome consistent with previous immunogold staining assays on endogenous Fwe (*Yao et al., 2009*). Upon high K$^+$ stimulation, SV and plasma membrane localizations were retained (*Figure 6—figure supplement 1d'*). Remarkably, DAB signals were apparent on all bulk endosomes (*Figure 6b–c*; *Figure 6—figure supplement 1d and d'*). To minimize staining variability across boutons, we compared DAB staining intensities on bulk endosomes and SVs in the same boutons. Compared to Flag-Fwe-HA rescue boutons, in HA-Fwe-APEX2 rescue boutons, the staining intensities of bulk endosomes were more abundant compared to those of the surrounding SVs (*Figure 6f*), revealing a mechanism by which Fwe is recycled to bulk endosomes after it initiates ADBE.

PI(4,5)P$_2$ is known to recruit adaptor protein complexes to sort SV proteins to the nascent SV during CME (*Saheki and De Camilli, 2012*). Next, we assessed if PI(4,5)P$_2$ microdomain formation may be involved in sorting of Fwe to bulk endosomes. When PI(4,5)P$_2$ microdomains were perturbed by Synj overexpression (*Figure 2b*), the bulk endosome localization of HA-Fwe-APEX2 was significantly

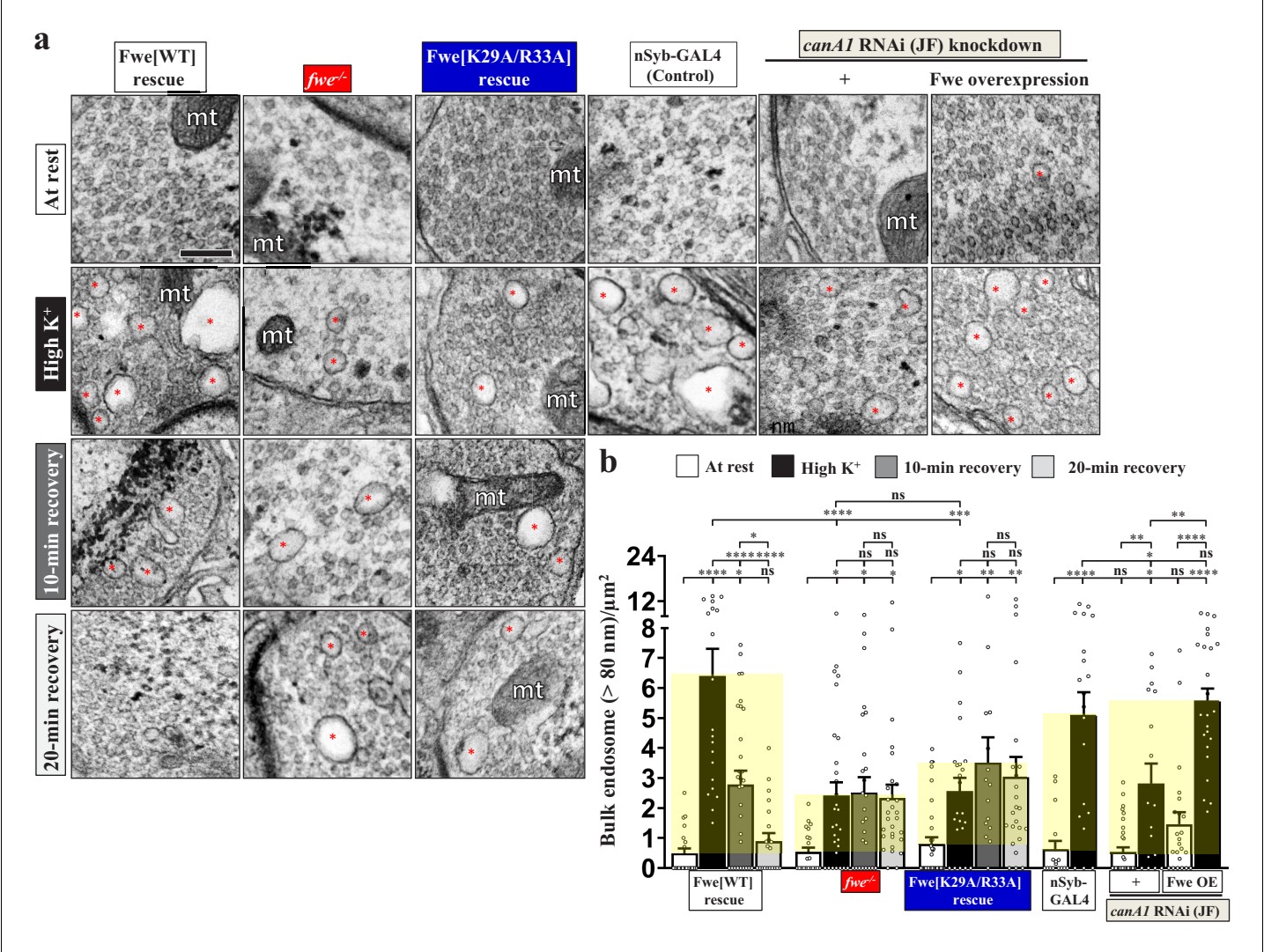

**Figure 5.** Perturbations of the positive feedback loop involving Fwe and PI(4,5)P$_2$ suppress ADBE and SV reformation from bulk endosomes. (a) TEM images of the boutons of HA-Fwe-APEX2 rescue (*nSyb-GAL4/UAS-HA-Fwe-APEX2 in fwe^{DB25/DB56}* at 25°C), *fwe* mutant (*fwe^{DB25/DB56}* at 25°C), HA-Fwe [K29A/R33A]-APEX2 rescue (*nSyb-GAL4/UAS-HA-Fwe[K29A/R33A]-APEX2 in fwe^{DB25/DB56}* at 25°C), control (*nSyb-GAL4* at 25°C), *canA1* RNAi (JF) knockdown (*nSyb-GAL4/UAS-canA1-RNAi (TRiP.JF01871)* at 29°C), *canA1* RNAi (JF) knockdown plus Fwe overexpression (*nSyb-GAL4/UAS-canA1-RNAi (TRiP.JF01871)/UAS-HA-Fwe-APEX2* at 29°C). TEM processing was performed after the following treatments: at rest (10 min incubation of 5 mM K$^+$/0 mM Ca$^{2+}$); high K$^+$ stimulation (10 min stimulation of 90 mM K$^+$/2 mM Ca$^{2+}$); 10 min recovery (10 min stimulation of 90 mM K$^+$/2 mM Ca$^{2+}$, followed by 10 min incubation of 5 mM K$^+$/0 mM Ca$^{2+}$); or 20 min recovery (10 min stimulation of 90 mM K$^+$/2 mM Ca$^{2+}$, followed by 20 min incubation of 5 mM K$^+$/0 mM Ca$^{2+}$). Bulk endosomes (>80 nm in diameter, red asterisks). Mitochondria (mt). (b) Quantification data of total numbers of bulk endosomes per bouton area. Individual data values are shown in graphs. p values: ns, not significant; *p<0.05; **p<0.01; ***p<0.001; ****p<0.0001. Mean ± SEM. Scale bar: 500 nm. Statistics: one-way ANOVA with Tukey's post hoc test.

The online version of this article includes the following source data and figure supplement(s) for figure 5:

**Source data 1.** Source data for *Figure 5*.
**Figure supplement 1.** Blockade of the positive feedback loop between Fwe and PI(4,5)P$_2$ abolishes SV reformation from the bulk endosome.
**Figure supplement 1—source data 1.** Source data for *Figure 5—figure supplement 1*.

reduced (*Figure 6d–f*), whereas there was only a mild reduction in its SV localization (*Figure 6d–e*). Therefore, in addition to initiating ADBE, PI(4,5)P$_2$ microdomains play a role in facilitating the retrieval of Fwe to the bulk endosome, enabling ADBE to remove its trigger via a negative feedback regulatory mechanism and reducing endocytosis to prevent excess membrane uptake.

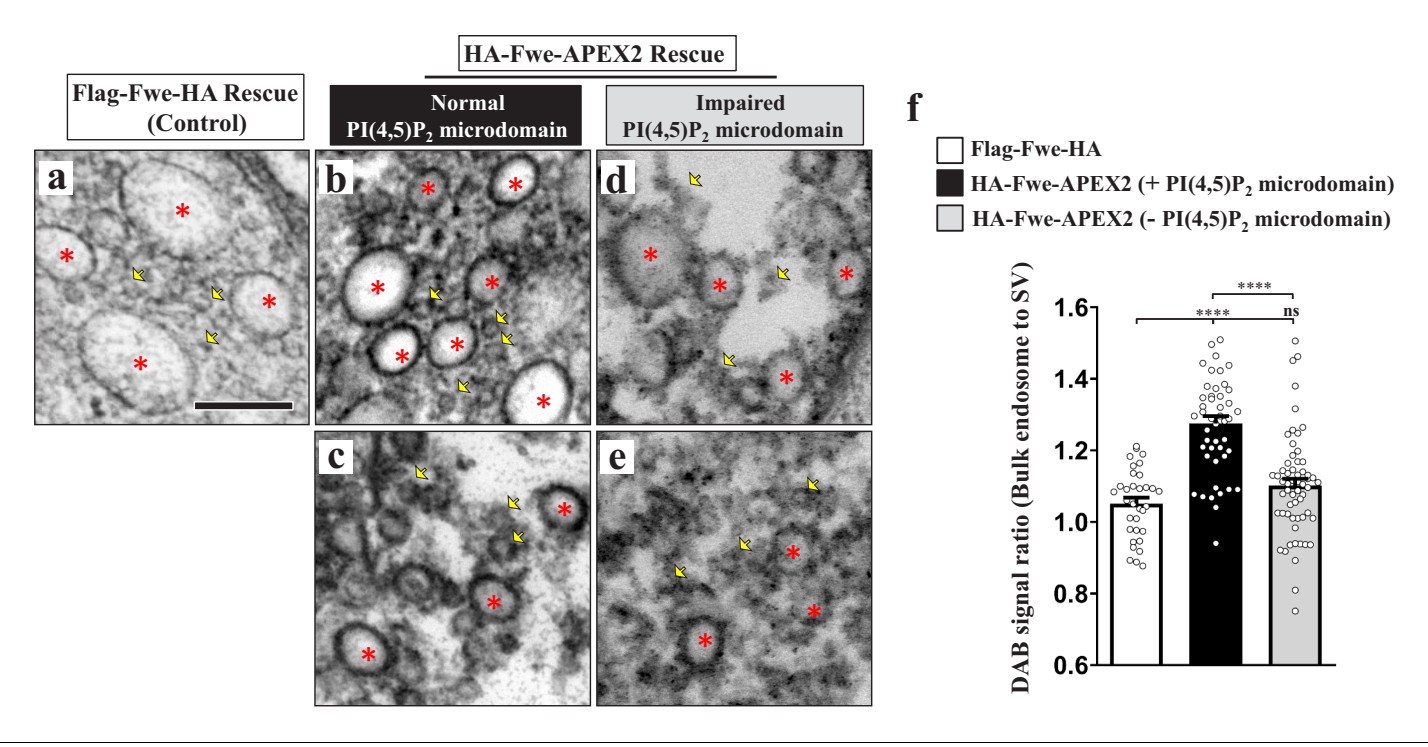

**Figure 6.** PI(4,5)P$_2$ facilitates recycling of Fwe to bulk endosomes. (a–f) PI(4,5)P$_2$ retrieves Fwe to bulk endosomes. (a–e) TEM images of the boutons of Flag-Fwe-HA rescue (*nSyb-GAL4/UAS-Flag-Fwe-HA in fwe$^{DB25/DB56}$* at 25°C, a), HA-Fwe-APEX2 rescue (*nSyb-GAL4/UAS-HA-Fwe-APEX2 in fwe$^{DB25/DB56}$* at 25°C, b–c), or HA-Fwe-APEX2 rescue coexpressing Synj (*nSyb-GAL4/UAS-HA-Fwe-APEX2/UAS-synj in fwe$^{DB25/DB56}$* at 25°C, d–e). After high K$^+$ stimulation, the boutons were subjected to DAB labeling. (a) The Flag-Fwe-HA rescue boutons presented DAB-negative bulk endosomes (red asterisks) and SVs (yellow arrows). (b and c) In the HA-Fwe-APEX2 rescue boutons, DAB signals on bulk endosomes (red asterisks) were higher than those on the SVs (yellow arrows). The views in b and c are from different boutons. (d and e) Under the condition of Synj overexpression, perturbation of PI(4,5)P$_2$ microdomain formation predominantly reduced DAB levels on the bulk endosomes (red asterisks). Views in d and e are from different boutons. (f) Quantification data for the DAB staining intensity ratio of bulk endosomes to surrounding SVs. The number of bulk endosomes, NMJ boutons, and larvae counted (Flag-Fwe-HA rescue control): Bulk endosomes (n = 33) derived from 5 NMJ boutons of two different larvae; HA-Fwe-APEX2 rescue: Bulk endosomes (n = 47) derived from 6 NMJ boutons of two different larvae; HA-Fwe-APEX2 rescue expressing Synj: Bulk endosomes (n = 59) derived from 9 NMJ boutons of two different larvae. Individual values were shown in graphs. p values: ns, not significant; ****p<0.0001. Mean ± SEM. Scale bar: 200 nm (a–e). Statistics: one-way ANOVA with Tukey's post hoc test.

The online version of this article includes the following source data and figure supplement(s) for figure 6:

**Source data 1.** Source data for *Figure 6*.
**Figure supplement 1.** HA-Fwe-APEX2 behaves like endogenous Fwe.

## Discussion

ADBE occurs immediately after exocytosis to retrieve required SV protein and lipid constituents to further regenerate SVs under conditions of high-frequency stimulations. Here, we show that the Fwe Ca$^{2+}$ channel-dependent compartmentalization of PI(4,5)P$_2$ orchestrates coupling of exocytosis to ADBE and subsequent SV reformation. Based on our findings, we propose a model for this interplay (depicted in *Figure 7*). Under conditions of strong stimulation, SV exocytosis transfers Fwe from SVs to the periactive zone, where some of the activated Fwe provides the low Ca$^{2+}$ levels that initiate Calcineurin activation to upregulate PI(4,5)P$_2$ (Step 1). Increased PI(4,5)P$_2$ enhances Fwe Ca$^{2+}$ channel activity, thereby establishing a positive feedback loop that induces PI(4,5)P$_2$ microdomain formation (Step 2). High levels of PI(4,5)P$_2$ within these microdomains elicit bulk membrane invagination by triggering actin polymerization (Step 3). In parallel, PI(4,5)P$_2$ facilitates proper retrieval of Fwe to the bulk endosome (Step 4), thereby terminating the ADBE process. Finally, PI(4,5)P$_2$ microdomains dictate SV reformation from the bulk endosomes (Step 5), coordinating ADBE and subsequent SV reformation.

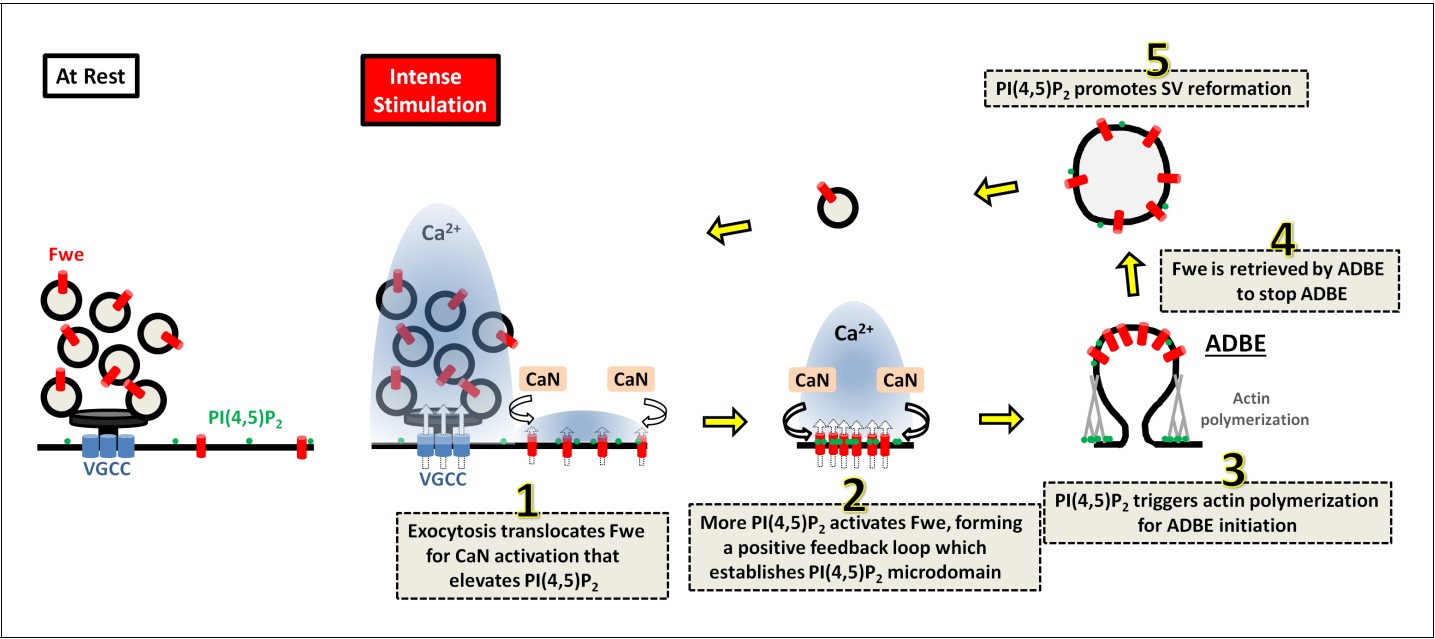

**Figure 7.** A proposed model for the role of Fwe-dependent PI(4,5)P$_2$ microdomains in coordinating ADBE and SV reformation from bulk endosomes. Details are described in the Discussion section.

## Fwe-dependent PI(4,5)P$_2$ microdomains trigger ADBE

The role of actin polymerization in ADBE has been investigated in mammals (*Kononenko et al., 2014*; *Soykan et al., 2017*; *Wu et al., 2016*), as well as in *Drosophila* (*Akbergenova and Bykhovskaia, 2009*). PI(4,5)P$_2$ is known to control a range of actin regulators, thereby modulating the dynamics of actin polymerization and branching (*Janmey et al., 2018*). It has been shown previously that, in response to nicotine stimulation, PI(4,5)P$_2$ forms clustered microdomains of sub-micrometer scale prior to the appearance of an actin-based ring structure in bovine chromaffin cells (*Gormal et al., 2015*). In agreement with this observation, we show that intense activity stimulation drives the formation of PI(4,5)P$_2$ microdomains at the periactive zone of *Drosophila* NMJ synaptic boutons. Perturbations of the formation of these microdomains reduces ADBE activity very significantly, demonstrating that rapid accumulation of PI(4,5)P$_2$ in microdomains is needed to trigger extensive actin polymerization, which likely generates sufficient mechanical force to produce the large endosomes. Furthermore, loss of *fwe* or RNAi-mediated calcineurin knockdown effectively inhibited PI(4,5)P$_2$ microdomain formation and, as a consequence, ADBE. Those results are consistent with previous data supporting that Ca$^{2+}$ promotes ADBE by activating its sensor Calcineurin (*Cousin and Robinson, 2001*; *Jin et al., 2019*; *Sun et al., 2010*; *Wu et al., 2009*; *Wu et al., 2014b*; *Xue et al., 2011*).

Our data provide evidence that Fwe-derived Ca$^{2+}$ regulates PI(4,5)P$_2$ dynamics through activation of Calcineurin. Phosphatidylinositol 4-phosphate 5-kinases (PIP5Ks) are the major kinases that promote PI(4,5)P$_2$ production from precursor phosphoinositides (*van den Bout and Divecha, 2009*). At mammalian central synapses, PIP5Kγ661, one of the PIP5Kγ isoforms, is the most abundant relative to other isoforms (*Wenk et al., 2001*). Activation of PIP5Ks can be engaged by binding to its regulators (*van den Bout and Divecha, 2009*). In response to Ca$^{2+}$ activation, Calcineurin has been shown to dephosphorylate PIP5Kγ661 to promote its interaction with AP-2 complexes, which augments the enzymatic activity of PIP5Kγ661 (*Nakano-Kobayashi et al., 2007*). Synj is the major PI(4,5)P$_2$ phosphatase in neurons (*Tsujishita et al., 2001*) and it is involved in the SV endocytosis prompted by strong stimulation (*Mani et al., 2007*). Whereas phosphorylation of Synj by CDK5 inhibits its activity, Calcineurin enhances Synj activity via dephosphorylation (*Lee et al., 2004*). In yeasts, cells undergo bulk membrane invagination, a process reminiscent of neuronal ADBE, under conditions of hyperosmotic stress (*Guiney et al., 2015*). However, in that process, Calcineurin dephosphorylates Synj to

alter its association with other endocytic partners rather than affecting its enzymatic activity to regulate PI(4,5)P$_2$ distribution. Intriguingly, divalent Ca$^{2+}$ ions are known to control the lateral organization of PI(4,5)P$_2$, further compartmentalizing PI(4,5)P$_2$ into ~70 nm-sized microdomains in monolayers of lipid bilayers via electrostatic interactions between Ca$^{2+}$ and PI(4,5)P$_2$ (*Carvalho et al., 2008*; *Ellenbroek et al., 2011*; *Levental et al., 2009*; *Sarmento et al., 2014*; *Wang et al., 2012*; *Wen et al., 2018*). Similarly-sized PI(4,5)P$_2$ clusters have been observed in PC12 cells under non-stimulated conditions (*van den Bogaart et al., 2011*). Therefore, tight spatial and temporal control of the localizations and activities of Fwe, Calcineurin, PIP5K, and Synj may drive Ca$^{2+}$-mediated PI(4,5)P$_2$ clustering, perhaps accounting for the Fwe-dependent formation of PI(4,5)P$_2$ microdomains at the periactive zone prior to ADBE. Further investigations are needed to characterize the underlying mechanisms.

Our data show that direct binding of PI(4,5)P$_2$ is required for the Ca$^{2+}$ channel activity of Fwe. Perturbation of PI(4,5)P$_2$-Fwe binding further impaired the formation of PI(4,5)P$_2$ microdomains as well as ADBE initiation. Hence, PI(4,5)P$_2$ controls Fwe gating, so that Fwe can promote PI(4,5)P$_2$ compartmentalization through positive feedback regulation. Furthermore, loss of Fwe impaired the intracellular Ca$^{2+}$ increase that was evoked upon strong activity stimulation (*Figure 3g*; *Yao et al., 2017*). These results support that, in addition to PI(4,5)P$_2$, the channel function of Fwe may be gated by a significant change in membrane potential. Expanding on that notion, it is therefore possible that both factors may gate Fwe, thereby only allowing channel opening when exocytosis directs Fwe to periactive zones. Future studies should explore the details of this channel gating mechanism.

## PI(4,5)P$_2$ microdomains coordinate retrieval of SV membranes and proteins for SV reformation via ADBE

Since ADBE is triggered very rapidly by intense stimuli, it was thought that this type of recycling randomly retrieves SV proteins and that the sorting process takes place when SVs regenerate from bulk endosomes. However, recent work has highlighted a distinct retrieval route for SV proteins during ADBE (*Kokotos et al., 2018*; *Nicholson-Fish et al., 2015*). Very little is known about the mechanisms underlying that retrieval route. Interestingly, removing VAMP4 or mutating its di-leucine motif was shown to impair ADBE (*Nicholson-Fish et al., 2015*). The di-leucine motif of transmembrane proteins is known to mediate binding with the AP-2 adaptor complex (*Traub and Bonifacino, 2013*). Given that the AP-2 adaptor complex works closely with PI(4,5)P$_2$ (*McMahon and Boucrot, 2011*), these findings imply a role for PI(4,5)P$_2$ and the AP-2 adaptor complex in SV protein sorting via ADBE. Indeed, our data show that bulk endosomes recycle few in a PI(4,5)P$_2$ microdomain-dependent manner. Hence, in addition to initiating ADBE, PI(4,5)P$_2$ may participate in SV protein sorting to bulk endosomes.

SV regeneration occurs following formation of the bulk endosome. Our results also show that either removing Fwe-derived Ca$^{2+}$ or perturbing PI(4,5)P$_2$ activity impaired the ability of SVs to reform from the bulk endosome, highlighting the essential role of PI(4,5)P$_2$ microdomains in this process. How could PI(4,5)P$_2$ of the plasma membrane regulate subsequent SV reformation? It has been shown that PI(4,5)P$_2$ is rapidly downregulated on bulk endosomes once formed by ADBE (*Chang-Ileto et al., 2011*; *Cremona et al., 1999*; *Milosevic et al., 2011*). It is conceivable that the high concentrations of PI(4,5)P$_2$ in microdomains may compensate for rapid turnover, thereby ensuring appropriate concentrations of PI(4,5)P$_2$ or PI(4)P for further recruitment of clathrin and adaptor protein complexes, such as AP-1 and AP-2 (*Blumstein et al., 2001*; *Cheung and Cousin, 2012*; *Faúndez et al., 1998*; *Glyvuk et al., 2010*; *Kokotos et al., 2018*; *Kononenko et al., 2014*; *Park et al., 2016*). Alternatively, PI(4,5)P$_2$ may facilitate SV protein sorting prior to ADBE, meaning proper SV protein compositions on bulk endosomes could control recruitment of adaptor protein complexes. Both of these potential mechanisms are not mutually exclusive and may operate in parallel. Therefore, we propose that the Fwe-dependent formation of PI(4,5)P$_2$ microdomains is potentially important in coordinating retrieval of SV membranes and cargos when SVs are recycled via ADBE. Notably, the Fwe channel is evolutionarily conserved from yeast to human (*Yao et al., 2009*). We have also previously demonstrated conserved functions of Fwe in CME and ADBE at the mammalian central synapse (*Yao et al., 2017*). A recent study has also identified Fwe as a key protein mediating Ca$^{2+}$-dependent granule endocytosis in mouse cytotoxic T lymphocytes (*Chang et al., 2018*). Hence, we hypothesize that the mechanism of ADBE we report here may be generally deployed across synapses and species, even in other non-neuronal cells.

## Materials and methods

### *Drosophila* strains and genetics

*fwe* mutants and transgenes: *fwe^DB25* and *fwe^DB56* (*Yao et al., 2009*); *UAS-Flag-Fwe-HA* (*Yao et al., 2009*); *UAS-Flag-Fwe[E79Q]-HA* (*Yao et al., 2017*); *UAS-canA1-RNAi(FB4)* (*Dijkers and O'Farrell, 2007*); *UAS-canA1-RNAi(TRiP.JF01871)* (Bloomington *Drosophila* Stock Center, BDSC#25850); *UAS-synj* (*Khuong et al., 2013*); *UAS-PLC_{δ1}-PH-EGFP* (*Verstreken et al., 2009*) (Bloomington *Drosophila* Stock Center, BDSC#39693); *UAS-PLC_{δ1}-PHS39R-EGFP* (*Verstreken et al., 2009*) (Bloomington *Drosophila* Stock Center, BDSC#39694); *nSyb-GAL4* (*Pauli et al., 2008*); *UAS-mCD8-EGFP* (Kyoto Stock Center, DGRC#108068); *vglut-lexA* (Bloomington *Drosophila* Stock Center, BDSC #60314); *13XLex-Aop2-IVS-GCaMP6f* (Bloomington *Drosophila* Stock Center, BDSC #44277)); *synj^1* (*Verstreken et al., 2003*) (Bloomington *Drosophila* Stock Center, BDSC#24883; *LexAop2-PLC_{δ1}-PH-APEX2-HA* (This paper); and *LexAop2-PLC_{δ1}-PH-EGFP* (This paper). Fly stocks were reared on regular food at 25°C or as otherwise indicated.

### Molecular cloning and transgenesis

For pUAST-HA-Fwe[WT]-APEX2, the coding sequence of the Fwe B isoform and APEX2 were separately PCR-amplified from pUAST-Flag-Fwe-HA (*Yao et al., 2009*) or pcDNA3 APEX2-NES (addgene #49386), respectively, and then subcloned into the pUAST vector. pUAST-HA-Fwe[K29A/R33A]-APEX2-HA was generated from pUAST-HA-Fwe[WT]-APEX2 through site-directed mutagenesis. pUAST-Flag-Fwe [K29/R33/K95/K100/R105/K146A/K147A/R150A]-HA, pUAST-Flag-Fwe[K146A/K147A/R150A]-HA and pUAST-Flag-Fwe[K95/K100/R105A]-HA were generated from pUAST-Flag-Fwe-HA through site-directed mutagenesis. For plexAop2-PLC_{δ1}-PH-APEX2-HA, the coding sequence of the PLC_{δ1}-PH domain and APEX2-HA were separately PCR-amplified and then subcloned into pJFRC19-13XLexAop2-IVS-myr:GFP (addgene #26224). For plexAop2-PLC_{δ1}-PH-EGFP, the coding sequence of PLC_{δ1}-PH-EGFP was PCR-amplified from genomic DNA of the *UAS-PLC_{δ1}-PHS39R-EGFP* fly stock and then subcloned into pJFRC19-13XLexAop2-IVS-myr:GFP. For YeMP-Nluc-Fwe-1D4, the cDNA fragment of Nluc and Fwe-1D4 was PCR-amplified from pNL1.1[Nluc] (a gift from Yi-Shiuan Huang) and YeMP-Fwe-1D4 plasmid (*Yao et al., 2009*), respectively, and then subcloned into the yeast expression YeMP vector. For YeMP-Nluc-Fwe[K29A/R33A]−1D4, YeMP-Nluc-Fwe[K95/K100/R105/K146/K147/R150A]−1D4, and YeMP-Nluc-Fwe[K29/R33/K95/K100/R105/K146/K147/R150A]−1D4, the DNA fragments of the Fwe variant were PCR-amplified from pUAST plasmids and sublconed into the YeMP-Nluc-Fwe-1D4 vector. PCR primers are indicated in key source table (Appendix 1). Transgenic flies were made by Wellgenetics Inc.

### Immunohistochemistry

Third instar larvae were fixed with 4% paraformaldehyde for 20 min. We used 1xPBS buffer containing 0.1% Tween-20 to stain the HA-tagged Fwe proteins. We used 1xPBS buffer containing 0.1% Triton X 100 to stain PLC_{δ1}-PH-EGFP or AP-2α. We used 1xPBS buffer containing 0.2% Triton X 100 to stain GCaMP6f. Primary antibodies were used as follows: guinea pig α-Fwe B isoform (1:400) (*Yao et al., 2017*), chicken α-GFP (Invitrogen, 1:500); mouse α-HA (Sigma, 1:400), mouse α-Bruchpilot (Developmental Studies Hybridoma Bank nc82, 1:100); rabbit α-AP-2α (1:3000) (*González-Gaitán and Jäckle, 1997*) guinea pig α-Eps15 (1:3000) (*Koh et al., 2007*); rabbit α-HRP conjugated with Alexa Fluor 488, Cy3 or Cy5 (Jackson ImmunoResearch Laboratories, 1:250). Secondary antibodies conjugated to Alexa Fluor 488, Alexa Fluor 555, or Alexa Fluor 647 (Invitrogen and Jackson ImmunoResearch) were used at 1:500. The NMJ boutons were derived from muscles 6 and 7 of abdominal segments 2/3. To detect $PI(4,5)P_2$ microdomains, the NMJ boutons were fixed immediately after high $K^+$ stimulation. To quantitatively compare PLC_{δ1}-PH-EGFP or AP-2α immunostaining signal among different sets of experiments, fixed larval fillets derived from different conditions were collected into the same Eppendorf tube. The NMJ boutons were stained for PLC_{δ1}-PH-EGFP using α-GFP and for the neuronal membranes using fluorescein-conjugated α-HRP. Consecutive single-plane images of the boutons of muscles 6 and 7 in abdominal segments 2 or 3 of all different experimental sets were taken using a Zeiss LSM 780 confocal microscope with a Plan-Apochromat 63x/1.4 Oil DIC M27 objective under a 1 μm interval setup and equal laser power and laser exposure time. For data quantification, single-plane images of five different individual boutons from each NMJ

bouton image were used. The presynaptic plasma membrane regions of the type Ib boutons defined by α-HRP immunostaining were outlined manually, and native fluorescence or α-GFP immunostaining signal intensities of $PLC_{\delta 1}$-PH-EGFP or AP-2α on the plasma membrane were quantified using ImageJ and averaged to serve as one individual data-point. For data quantification of SIM images, we chose single-plane SIM images focused on a central section of an individual bouton and outlined plasma membrane-associated $PLC_{\delta 1}$-PH-EGFP-immunostained clusters using ImageJ. The area of these clusters of each individual bouton was measured using ImageJ, and the areas of clusters measuring over 0.032 $\mu m^2$ (equal to a ~ 200 nm diameter circle, i.e. the resolution limit of SIM) were averaged to serve as one data-point. We assessed 37 boutons derived from five NMJs of three different larvae. Image processing was achieved using LSM Zen.

## Western blot

For western blotting, the brain and ventral nerve chord of larval fillets were removed and subjected to different stimulation conditions. Afterwards, the fillets were crushed in 1xSDS sample buffer and boiled for 5 min. Dilutions for primary antibodies were as follows: mouse anti-α-actin, 1:20000 (Sigma); chicken anti-GFP, 1:5000 (Invitrogen).

## PLA

Third-instar larvae were fixed with 4% paraformaldehyde for 20 min and permeabilized with 1xPBS buffer containing 0.1% Tween-20. Larval fillets were incubated with mouse α-HA (Sigma, 1:200) and rabbit α-GFP (Invitrogen, 1:500) in 1xPBS buffer containing 0.1% Tween-20 at 4°C for 12 hr. Excess antibodies were washed out using 1xPBS buffer containing 0.1% Tween-20. The samples were mixed with the PLA probe (Sigma, 1:5) for 2 hr at 37°C. After washing with 1x buffer A, the samples were incubated with ligation solution (1:40) for 1.5 hr at 37°C. After again washing with 1x buffer A, the samples were incubated with amplification solution (1:80) for 2 hr at 37°C. Next, the samples were washed with 1x buffer B and then 0.01x buffer B. The samples were stained with anti-chicken Alexa Fluor 488-conjugated IgG and anti-mouse Alexa Fluor 647-conjugated IgG, followed by a wash of 1x PBS buffer containing 0.1% Tween-20. To quantitatively compare PLA signal, fixed larval fillets derived from different experimental conditions were collected into the same Eppendorf tube and processed. Consecutive single-plane images of the boutons of muscles 6 and 7 in abdominal segments 2 or 3 of all different experimental sets were taken using a Zeiss LSM 780 confocal microscope with a Plan-Apochromat 63x/1.4 Oil DIC M27 objective under a 1 μm interval setup and equal laser power and laser exposure time. For data quantification, consecutive Z-plane images spanning whole NMJ were projected under maximal fluorescence intensity. All type Ib boutons in individual Z-projection image were outlined according to $PLC_{\delta 1}$-PH-EGFP-stained regions. PLA or antibody immunostaining signal intensities within the boutons and background staining signals in surrounding muscles were counted using ImageJ and averaged. One individual data-point was obtained by muscular background signal subtraction. Image processing was achieved using LSM Zen.

## Nluc-Fwe-1D4 purification and BRET assay

The Nluc-Fwe-1D4 fusion protein was purified as described previously (*Yao et al., 2009*). Briefly, plasma membrane was isolated from a two liter culture of yeast strain BJ5457 expressing Nluc-Fwe-1D4 protein. The plasma membranes were solubilized at 4°C with 10x critical micelle concentration (CMC) DDM (Anatrace) in a solution of 20 mM HEPES (pH8.0), 300 mM NaCl, 10% Glycerol, 2.0 mM DTT, and 1 mM PMSF for 2 hr. Insoluble membranes were spun down by centrifugation at 100,000 $\times g$ for 60 min. The lysates including solubilized Nluc-Fwe-1D4 proteins were cleaned with CNBr sepharose 4B at 4°C for 1 hr. The samples were then mixed with α−1D4-conjugated CNBr-activated Sepharose 4B at 4°C for 8–12 hr. After washing with a solution of 20 mM HEPES (pH8.0), 150 mM NaCl, 10% Glycerol, 2.0 mM DTT, 1 mM PMSF and 2.6xCMC DDM, the protein was eluted with a 1D4 peptide-containing buffer (3 mg of 1D4 peptide in 1 ml of washing buffer). Purified proteins were subjected to SDS-PAGE and detected using Lumitein staining or western blotting with anti-α−1D4 antibody at 1:5000 (*Yao et al., 2009*). The BRET assay was performed in a 384-well plate, with each well containing 30 μl of reaction solution [0.5 nM purified proteins, 5 μM BODIPY-TMR Phosphatidylinositol 4,5-bisphosphate (C-45M16A, Echelon Bioscience), furimazine (Promega, 1:2000), 20 mM HEPES, 150 mM NaCl, 10% Glycerol, 2 mM DTT, 1 mM PMSF, 4 mM DDM].

Fluorescence signal was detected using a Microplate Reader M1000 pro (Tecan) with two different emission spectrum filters, that is 500–540 nm for Nluc and 550–630 nm for BODIPY-TMR Phosphatidylinositol 4,5-bisphosphate. For competition assay, 30 µl of the reaction solution was included with 1 mM brain phosphatidylinositol 4,5-bisphosphate (Avanti). The BRET signal was calculated according to the following formula:

## Live imaging

For PLC$_{\delta1}$-PH-EGFP imaging, third-instar larvae were dissected in a zero-calcium HL-3 solution at room temperature. For groups stimulated with electric pulses, larval fillets were bathed in a solution of 2 mM Ca$^{2+}$ (70 mM NaCl, 5 mM KCl, 10 mM MgCl$_2$, 10 mM NaHCO$_3$, 5 mM trehalose, 5 mM HEPES (pH 7.4), 115 mM sucrose, 2 mM CaCl$_2$). High concentrations of glutamate were used to desensitize glutamate receptors, thereby reducing muscle contraction when stimulated. A cut axonal bundle was sucked into the tip of a glass capillary electrode and then stimulated at 20 or 40 Hz for 3 min. Stimulus strength was set at 5 V and 0.5 ms duration by means of pClamp 10.6 software (Axon Instruments Inc). Ten images were taken from larval fillets at rest. After stimulation for 2 min, muscle contraction significantly decelerated. Thus, we captured 60 consecutive snapshot images every second from the third minute. Muscles 6 and 7 of abdominal segment three were imaged. Under the condition of high K$^+$ stimulation, larval fillets were bathed in a solution of 90 mM K$^+$/2 mM Ca$^{2+}$/7 mM glutamate (25 mM NaCl, 90 mM KCl, 10 mM MgCl$_2$, 10 mM NaHCO$_3$, 5 mM trehalose, 5 mM HEPES (pH 7.4), 30 mM sucrose, 2 mM CaCl$_2$, 7 mM monosodium glutamate) for 5 min. Sixty consecutive snapshot images were captured every second from the fifth minute of stimulation. Images were taken using a long working distance water immersion objective (XLUMPLFLN20XW, Olympus) and EMCCD camera (iXon, Andor) mounted on a SliceScope Pro 6000 (Scientifica) microscope and employing MetaFluor software (Molecular Devices). For GCaMP6f imaging, third instar larvae were dissected in a zero-calcium HL-3 solution at room temperature. Ten images were taken from larval fillets at rest. Subsequently, larval fillets were stimulated with a solution of 90 mM K$^+$/2 mM Ca$^{2+}$/7 mM glutamate for 5 min. Sixty consecutive snapshot images were captured every second from the fifth minute of stimulation. The *lexA/lexAop2* binary system was used to stably express a comparable level of GCaMP6f in the presynaptic compartment of NMJ boutons for all tested genotypes, allowing us to compare Ca$^{2+}$ imaging results when the resting Ca$^{2+}$ levels were potentially affected by differences in genetic background. We stained for the GCaMP6f protein using a-GFP antibody and confirmed comparable GCaMP6f levels among the different genotypes tested in each dataset. Evoked Ca$^{2+}$ levels were calculated by subtracting the resting GCaMP6f fluorescence from the GCaMP6f fluorescence induced by high K$^+$ stimulation. The NMJs were derived from muscles 6 and 7 of abdominal segment 2/3. Images were taken using a water immersion objective (W Plan-Apochromat 40x/1.0 DIC M27, Zeiss). For each imaging experiment, at least three focused images for the same boutons under resting, stimulation, or post-stimulation conditions were used for data quantification. Fluorescence intensities of PLC$_{\delta1}$-PH-EGFP or GCaMP6f within the boutons were quantified using ImageJ and averaged to serve as one individual data-point. Image processing was achieved using LSM Zen.

## Transmission electron microscopy

Third instar larval fillets were prepared in zero-calcium HL-3 solution at room temperature. For the resting conditions, the fillets were bathed in zero-calcium HL-3 solution at room temperature for another 10 min before fixation. For the high K$^+$ stimulation conditions, fillets were bathed in a solution of 90 mM K$^+$ and 2 mM Ca$^{2+}$ for 10 min. The stimulation was terminated by washing three times with zero-calcium HL-3 solution, followed by fixation. For recovery conditions, following high K$^+$ stimulation, fillets were bathed in zero-calcium HL-3 solution at room temperature for 10 or 20 min before fixation. Larval fillets were fixed for 12 hr at 4°C in 4% paraformaldehyde/1% glutaraldehyde/0.1 M cacodylic acid (pH 7.2), rinsed with 0.1 M cacodylic acid (pH 7.2), and postfixed with 1% OsO$_4$ and 0.1 M cacodylic acid at room temperature for 3 hr. These samples were then subjected to a series of dehydration steps using 30–100% ethanol. After 100% ethanol dehydration, the samples were sequentially incubated with propylene, a mixture of propylene and resin, and pure resin. Finally, the samples were embedded in 100% resin. TEM images were captured using Tecnai G2 Spirit TWIN (FEI Company) and a Gatan CCD Camera (794.10.BP2MultiScanTM). NMJ boutons were

captured at high magnifications. For each condition, NMJ bouton images were taken from at least five different NMJs of each third-instar larvae, and three to five larvae were used. Quantifications were performed using ImageJ. For diaminobenzidine (DAB) polymerization, third instar larvae were dissected at room temperature in zero-calcium HL-3 medium, followed by a 10 min incubation in 5 mM $K^+$/0 $Ca^{2+}$ mM solution or a 10 min stimulation of 90 mM $K^+$/2 mM $Ca^{2+}$. Next, the samples were subjected to 30 min fixation in ice-cold 4% paraformaldehyde/1% glutaraldehyde/0.1 M cacodylic acid (pH 7.2). Subsequently, the samples were transferred to Eppendorf tubes for 15 min incubation with a solution of 0.5 mg/ml DAB solution, followed by incubation with a solution of 0.5 mg/ml DAB and 0.006% $H_2O_2$ for 15 min at room temperature. This latter step was repeated once to ensure DAB polymerization. Samples were washed three times with 1xPBS buffer for 10 min and then fixed with a solution of 4% paraformaldehyde/1% glutaraldehyde/0.1 M cacodylic acid (pH 7.2) for 12 hr at 4°C, followed by fixation with a solution of 1% $OsO_4$/0.1 M cacodylic acid at room temperature for 3 hr. Then, standard dehydration, embedding, and imaging were performed. For data quantifications of DAB intensities, the display color of TEM images was reverted to grayscale using ImageJ. Average DAB staining intensity on each individual bulk endosome was quantified. Then, the average DAB staining intensity on 50–100 surrounding SVs from the same bouton image was used to assess the relative level of HA-Fwe-APEX2 on bulk endosomes vs SVs.

## Statistics

All data analyses were conducted using GraphPad Prism 8.0, unless stated otherwise. Paired and multiple datasets were compared by Student $t$-test or one-way ANOVA with Tukey's post hoc test, respectively. Individual data values are biological replicates. Samples were randomized during preparation, imaging, and data processing to minimize bias.

## Acknowledgements

We thank Hugo Bellen, Patrik Verstreken, Kartik Venkatachalam, the Bloomington *Drosophila* Stock Center, and the Developmental Studies Hybridoma Bank for stocks and reagents. We thank Hugo Bellen, Ruey-Hwa Chen, and Y Henry Sun for the critical comments. We thank Yi-Shuian Huang for providing the Nluc plasmid. We thank DNA Sequencing Core Facility (AS-CFII-108–115) for sequencing DNA constructs. We thank Wellgenetics for making transgenic lines. We thank the IMB imaging core for helping with TEM and SIM imaging. This work was supported by grants from the Ministry of Science and Technology of Taiwan (107–2311-B-001–003-MY3, 106-0210-01-15-02, and 107-0210-01-19-01).

## Additional information

### Funding

| Funder | Grant reference number | Author |
| --- | --- | --- |
| Ministry of Science and Technology, Taiwan | 107-2311-B-001-003-MY3 | Chi-Kuang Yao |
| Ministry of Science and Technology, Taiwan | 106-0210-01-15-02 | Chi-Kuang Yao |
| Ministry of Science and Technology, Taiwan | 107-0210-01-19-01 | Chi-Kuang Yao |

The funders had no role in study design, data collection and interpretation, or the decision to submit the work for publication.

### Author contributions

Tsai-Ning Li, Yu-Jung Chen, Conceptualization, Resources, Data curation, Software, Formal analysis, Validation, Investigation, Visualization, Methodology, Project administration, Writing - review and editing; Ting-Yi Lu, Data curation, Formal analysis, Investigation, Methodology; You-Tung Wang, Hsin-Chieh Lin, Resources, Data curation, Software, Formal analysis, Validation, Investigation, Visualization, Methodology; Chi-Kuang Yao, Conceptualization, Resources, Data curation, Software, Formal

analysis, Supervision, Funding acquisition, Validation, Investigation, Visualization, Methodology, Writing - original draft, Project administration, Writing - review and editing

### Author ORCIDs
Tsai-Ning Li (iD) https://orcid.org/0000-0003-0195-145X
Chi-Kuang Yao (iD) https://orcid.org/0000-0003-0977-4347

### Decision letter and Author response
Decision letter https://doi.org/10.7554/eLife.60125.sa1
Author response https://doi.org/10.7554/eLife.60125.sa2

## Additional files
### Supplementary files
- Transparent reporting form

### Data availability
All data generated or analysed during this study are included in the manuscript and supporting files. Source data files have been provided.

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

# Appendix 1

## key resources table

**Appendix 1—key resources table**

| Reagent type (species) or resource | Designation | Source or reference | Identifiers | Additional information |
|---|---|---|---|---|
| Genetic reagent (*D. melanogaster*) | *fwe*[DB25] | *Yao et al., 2009* | | *fwe* mutant allele used in *Figures 3–6*, *Figure 4—figure supplements 1–2*, *Figure 5—figure supplement 1*, and *Figure 6—figure supplement 1* |
| Genetic reagent (*D. melanogaster*) | *fwe*[DB56] | *Yao et al., 2009* | | *fwe* mutant allele used in *Figures 3–6*, *Figure 4—figure supplements 1–2*, *Figure 5—figure supplement 1*, and *Figure 6—figure supplement 1* |
| Genetic reagent (*D. melanogaster*) | *UAS-Flag-Fwe-HA* | *Yao et al., 2017* | | Wild-type version of *fwe* transgene used in *Figure 3a–b*, *Figure 6a*, *Figure 4—figure supplement 1*, and *Figure 6—figure supplement 1b* |
| Genetic reagent (*D. melanogaster*) | *UAS-Flag-Fwe[E79Q]-HA* | *Yao et al., 2017* | | Channel deficient version of *fwe* transgene used in *Figure 3e–f* |
| Genetic reagent (*D. melanogaster*) | *UAS-canA1-RNAi (FB4)* | *Dijkers and O'Farrell, 2007* | | RNAi line of *canA1* used in *Figure 3g–j* |
| Genetic reagent (*D. melanogaster*) | *UAS-canA1-RNAi(TRiP. JF01871)* | Bloomington *Drosophila* Stock Center *Dijkers and O'Farrell, 2007* | RRID:BDSC_25850 | RNAi line of *canA1* used in *Figure 3g–h*, 5 |
| Genetic reagent (*D. melanogaster*) | *UAS-synj* | *Khuong et al., 2013* | | Wild-type version of *synj* transgene used in *Figure 2b–d*, *Figure 3i–j*, *Figure 6d–f*, and *Figure 2—figure supplement 1c*. |
| Genetic reagent (*D. melanogaster*) | *UAS-PLC$_{\delta 1}$-PH-EGFP* | Bloomington *Drosophila* Stock Center *Verstreken et al., 2009* | RRID:BDSC_39693 | PI(4,5)P$_2$ reporter transgene used in *Figures 1–4*, *Figure 1—figure supplement 1*, *Figure 1—figure supplement 3c*, *Figure 2—figure supplements 1–2*, *Figure 4—figure supplement 1e–f*. |
| Genetic reagent (*D. melanogaster*) | *UAS-PLC$_{\delta 1}$-PHS39R-EGFP* | Bloomington *Drosophila* Stock Center *Verstreken et al., 2009* | RRID:BDSC_39694 | PI(4,5)P$_2$-binding mutant transgene used in *Figure 2c–d*, *Figure 1—figure supplement 2c–e* |

*Continued on next page*

*Appendix 1—key resources table continued*

| Reagent type (species) or resource | Designation | Source or reference | Identifiers | Additional information |
|---|---|---|---|---|
| Genetic reagent (*D. melanogaster*) | nSyb-GAL4 | Bloomington *Drosophila* Stock Center **Pauli et al., 2008** | RRID:BDSC_51635 | Neuronal *GAL4* driver used in all figures |
| Genetic reagent (*D. melanogaster*) | vglut-lexA | Bloomington *Drosophila* Stock Center | RRID:BDSC_60314 | Motor neuron *lexA* driver used in *Figure 3e–f*, *Figure 3i–j*, *Figure 4d–e*, and *Figure 4—figure supplement 1c–d* |
| Genetic reagent (*D. melanogaster*) | 13XLexAop2-IVS-GCaMP6f | Bloomington *Drosophila* Stock Center | RRID:BDSC_44277 | $Ca^{2+}$ indicator transgene used in *Figure 3i–j*, *Figure 4d–e*, and *Figure 4—figure supplement 1c–d* |
| Genetic reagent (*D. melanogaster*) | synj[1] | Bloomington *Drosophila* Stock Center **Verstreken et al., 2003** | RRID:BDSC_24883 | *synj* mutant allele used in *Figure 1—figure supplement 1c–d* |
| Genetic reagent (*D. melanogaster*) | UAS-mCD8-EGFP | Kyoto Stock Center | RRID:DGRC_108068 | *mCD8-EGFP* transgene used in *Figure 1—figure supplement 2a–b* |
| Genetic reagent (*D. melanogaster*) | UAS-HA-Fwe[WT]-APEX2 | This paper | | APEX2 fusion version of wild-type *fwe* transgene used in *Figure 4d–g*, *Figure 5*, *Figure 6b–f*, *Figure 4—figure supplement 2*, and *Figure 6—figure supplement 1c–d* |
| Genetic reagent (*D. melanogaster*) | UAS-HA-Fwe[K29A/R33A]-APEX2 | This paper | | APEX2 fusion version of *fwe [K29A/R33A]* transgene used in *Figure 4d–g*, *Figure 5*, *Figure 4—figure supplement 2* |
| Genetic reagent (*D. melanogaster*) | UAS-Flag-Fwe[K29A/R33A/K95A/K100A/R105A/K146A/K147A/R150A]-HA | This paper | | fwe[K29A/R33A/K95A/K100A/R105A/K146A/K147A/R150A] transgene was not expressed stably |
| Genetic reagent (*D. melanogaster*) | UAS-Flag-Fwe[K146A/K147A/R150A]-HA | This paper | | fwe[K146A/K147A/R150A] transgene used in *Figure 4—figure supplement 1* |
| Genetic reagent (*D. melanogaster*) | UAS-Flag-Fwe[K95A/K100A/R105A]-HA | This paper | | fwe[K95A/K100A/R105A] transgene was not expressed stably |
| Genetic reagent (*D. melanogaster*) | LexAop2-PLC$_{\delta 1}$-PH-APEX2-HA | This paper | | APEX2 fusion version of PLCδ1-PH transgene used in *Figure 3i–j* |
| Genetic reagent (*D. melanogaster*) | LexAop2-PLCδ1-PH-EGFP | This paper | | *lexAop2* version of PLC$_{\delta 1}$-PH-EGFP transgene used in *Figure 3e–f* |
| Antibody | α-GFP (Chicken polyclonal) | Invitrogen | Cat #A10262, RRID:AB_2534023 | IF: 1:500 WB: 1:5000 |
| Antibody | α-HA (Mouse monoclonal) | Sigma | Cat # H3663, RRID:AB_262051 | IF: 1:400 PLA: 1:200 |

*Continued on next page*

*Appendix 1—key resources table continued*

| Reagent type (species) or resource | Designation | Source or reference | Identifiers | Additional information |
|---|---|---|---|---|
| Antibody | α-Bruchpilot (Mouse monoclonal) | DSHB *Wagh et al., 2006* | Cat# nc82, RRID:AB_2314866 | IF: 1:100 |
| Antibody | α-AP-2α (Rabbit polyclonal) | *González-Gaitán and Jäckle, 1997* | | IF: 1:3000 |
| Antibody | α-Eps15 (Guinea pig polyclonal) | *Koh et al., 2007* | | IF: 1:3000 |
| Antibody | α-Fwe B isoform (Guinea pig polyclonal) | *Yao et al., 2017* | | IF: 1:400 |
| Antibody | Cy3 AffiniPure Rabbit Anti-Horseradish Peroxidase | Jackson Immuno Research Labs | Cat# 323-165-021, RRID:AB_2340262 | IF: 1:250 |
| Antibody | Alexa Fluor 488 AffiniPure Rabbit Anti-Horseradish Peroxidase | Jackson Immuno Research Labs | Cat# 323-545-021, RRID:AB_2340264 | IF: 1:250 |
| Antibody | α-GFP (Rabbit polyclonal) | Thermo Fisher Scientific | Cat# A-6455, RRID: AB_221570 | PLA: 1:500 |
| Antibody | α-actin (Mouse monoclonal) | Sigma | Cat# A7732, RRID: AB_2221571 | WB:1:20000 |
| Antibody | α−1D4 (Mouse monoclonal) | *Yao et al., 2009* | | WB: 1:2000 |
| Strain, strain background (*Escherichia coli*) | DH5α | | | |
| Strain, strain background (*Escherichia coli*) | yeast strain BJ5457 | *Yao et al., 2009* | | |
| Peptide, recombinant protein | Nluc-Fwe-1D4 fusion protein | This paper | | Used in *Figure 4b–c* |
| Peptide, recombinant protein | Nluc-Fwe[K29A/R33A]−1D4 | This paper | | Used in *Figure 4b–c* |
| Peptide, recombinant protein | Nluc-Fwe[K95A/K100A/R105A/K146A/K147/R150A]−1D4 | This paper | | Used in *Figure 4b–c* |
| Peptide, recombinant protein | Nluc-Fwe[K29A/R33A/K95A/K100A/R105A/K146A/K147A/R150A]−1D4 | This paper | | Used in *Figure 4b–c* |
| Chemical compound, drug | n-Dodecyl-β-D-Maltopyranoside | Anatrace | Cat# D310 | Used in *Figure 4b–c* |
| Chemical compound, drug | BODIPY-TMR Phosphatidylinositol 4,5-bisphosphate | Echelon Bioscience | Cat#C-45M16A | Used in *Figure 4b–c* |
| Chemical compound, drug | Furimazine | Promega | Cat# N1110 | Used in *Figure 4b–c* |

*Continued on next page*

*Appendix 1—key resources table continued*

| Reagent type (species) or resource | Designation | Source or reference | Identifiers | Additional information |
|---|---|---|---|---|
| Chemical compound, drug | Brain Phosphatidylinositol 4,5-bisphosphate | Avanti Polar Lipids | AV-840046P | Used in *Figure 4b–c* |
| Commercial assay or kit | Duolink In Situ Red Starter Kit Mouse/Rabbit | Sigma | Cat# DUO92101 | Used in *Figure 3a–b* |
| Commercial assay or kit | NEBuilder HiFi DNA Assembly Master Mix | NEB | Cat# E2621 | |
| Recombinant DNA reagent | YeMP | *Yao et al., 2009* | | |
| Recombinant DNA reagent | pcDNA3 APEX2-NES | Addgene | RRID:Addgene_49386 | |
| Recombinant DNA reagent | pJFRC19-13XLexAop2-IVS-myr:GFP | Addgene | RRID:Addgene_26224 | |
| Recombinant DNA reagent | pNL1.1[Nluc] | Promega | Cat# #N1001 | |
| Recombinant DNA reagent | pUAST-HA-Fwe[WT]-APEX2 | This paper | | DNA construct of APEX2 fusion version of wild-type fwe transgene used in *Figure 4d–g*, *Figure 5*, *Figure 6b–f*, *Figure 4—figure supplement 2*, and *Figure 6—figure supplement 1c–d* |
| Recombinant DNA reagent | pUAST-HA-Fwe[K29A/R33A]-APEX2 | This paper | | DNA construct of APEX2 fusion version of fwe[K29A/R33A] transgene used in *Figure 4d–g*, *Figure 5*, *Figure 4—figure supplement 2* |
| Recombinant DNA reagent | pUAST-Flag-Fwe-HA | *Yao et al., 2017* | | |
| Recombinant DNA reagent | pUAST-Flag-Fwe[K29/R33/K95/K100/R105/K146A/K147A/R150A]-HA | This paper | | DNA construct of fwe[K29A/R33A/K95A/K100A/R105A/K146A/K147A/R150A] transgene |
| Recombinant DNA reagent | pUAST-Flag-Fwe[K146A/K147A/R150A]-HA | This paper | | DNA construct of fwe[K146A/K147A/R150A] transgene used in *Figure 4—figure supplement 1* |
| Recombinant DNA reagent | pUAST-Flag-Fwe[K95/K100/R105A]-HA | This paper | | DNA construct of fwe[K95A/K100A/R105A] transgene |
| Recombinant DNA reagent | plexAop2-PLC$_{\delta 1}$-PH-APEX2-HA | This paper | | DNA construct of APEX2 fusion version of PLCδ1-PH transgene used in *Figure 3i–j* |
| Recombinant DNA reagent | plexAop2-PLC$_{\delta 1}$-PH-EGFP | This paper | | DNA construct of lexAop2 version of PLC$_{\delta 1}$-PH-EGFP transgene used in *Figure 3e–f* |

*Continued on next page*

*Appendix 1—key resources table continued*

| Reagent type (species) or resource | Designation | Source or reference | Identifiers | Additional information |
|---|---|---|---|---|
| Recombinant DNA reagent | YeMP-Nluc-Fwe-1D4 | This paper | | DNA construct of expression of Nluc-Fwe-1D4 recombinant protein used in *Figure 4b–c* |
| Recombinant DNA reagent | YeMP-Nluc-Fwe[K29A/R33A]—1D4 | This paper | | DNA construct of expression of Nluc-Fwe[K29A/R33A]—1D4 recombinant protein used in *Figure 4b–c* |
| Recombinant DNA reagent | YeMP-Nluc-Fwe[K95A/K100A/R105A/K146A/K147/R150A]—1D4 | This paper | | DNA construct of expression of Nluc-Fwe[K95A/K100A/R105A/K146A/K147/R150A]—1D4 recombinant protein used in *Figure 4b–c* |
| Recombinant DNA reagent | YeMP-Nluc-Fwe[K29A/R33A/K95A/K100A/R105A/K146A/K147A/R150A]—1D4 | This paper | | DNA construct of expression of Nluc-Fwe[K29A/R33A/K95A/K100A/R105A/K146A/K147A/R150A]—1D4-1D4 recombinant protein used in *Figure 4b–c* |
| Sequence-based reagent | HA-Fwe[WT]-APEX2 forward-1 | This paper | | 5'-tcgttaacagatcttGCGGCCGCAT GTACCCATACGATG TTCCAGATTACG-3' |
| Sequence-based reagent | HA-Fwe[WT]-APEX2 Reverse-1 | This paper | | 5'- tgaacctcctccgccTGT GGGGCGCCAGACATC-3' |
| Sequence-based reagent | HA-Fwe[WT]-APEX2 forward-2 | This paper | | 5'- gccccacaggcggaggaggttca ggaggcggaggttcgGG TACCGG CGGAAAGAGTTACCCGAC-3' |
| Sequence-based reagent | HA-Fwe[WT]-APEX2 Reverse-2 | This paper | | 5'- tccttcacaaagatccTCTAGAG GCGTCGGCGAATCCCAG —3' |
| Sequence-based reagent | HA-Fwe[K29A/R33A]-APEX2-HA forward | This paper | | 5'-CAGCCGTGGTATC TCGCA TATGGAAGTGCATTGC TGGGCATTG-3' |
| Sequence-based reagent | Flag-Fwe[K95/K100/R105A]-HA forward | This paper | | 5'-GGAATCTGCGCCCTTG TACTT CGCTGCCGGGCTCTACA TTGCC ATGGCCATTCCGCCCATTA TT-3' |
| Sequence-based reagent | Flag-Fwe[K146A/K147A/R150A]-HA forward | This paper | | 5'-CAGAGGACA TGGCCGCCGC TGCCACA TCGCCCACACAGATG GCCGGCAG TCAGGCGGGCGG-3' |
| Sequence-based reagent | PLC$_{\delta 1}$-PH-APEX2-HA forward-1 | This paper | | 5'-tcttatcctttacttcagGCGGC CGCATGGAC TCGGGCCGGGAC-3' |

*Continued on next page*

*Appendix 1—key resources table continued*

| Reagent type (species) or resource | Designation | Source or reference | Identifiers | Additional information |
|---|---|---|---|---|
| Sequence-based reagent | PLC$_{\delta 1}$-PH-APEX2-HA forward-2 | This paper | | 5'- GCTGTACAAG ggcggaggaggttca ggaggcggaggttcgGGCGGAA AGAGTTACCCGAC-3' |
| Sequence-based reagent | PLC$_{\delta 1}$-PH-APEX2-HA reverse-1 | This paper | | 5'- cctccgccggtaccAAGATCT TCCGGGCATAGCTGTCG-3' |
| Sequence-based reagent | PLC$_{\delta 1}$-PH-APEX2-HA reverse-2 | This paper | | 5'- ggttccttcacaaagatcctctagattaa gcgtaatctggaacatcgtatgggtaG GCGTCGGCGAATCCCAG −3' |
| Sequence-based reagent | PLC$_{\delta 1}$-PH-EGFP forward | This paper | | 5'- tcttatcctttacttcagGCGGCC GCATGGAC TCGGGCCGGGAC-3' |
| Sequence-based reagent | PLC$_{\delta 1}$-PH-EGFP reverse | This paper | | 5'- TTTCTAGATTACTTG TACAGCTCGTCCAT-3' |
| Sequence-based reagent | YeMP-Nluc-Fwe-1D4 forward-1 | This paper | | 5'- CCACTAGTATGGT CTTCACACTCGAAG-3' |
| Sequence-based reagent | YeMP-Nluc-Fwe-1D4 forward-2 | This paper | | 5'- AAAACTAGTGTCGAC TCGT TTGCGGAAAAGATAACG −3' |
| Sequence-based reagent | YeMP-Nluc-Fwe-1D4 Reverse-1 | This paper | | 5'-AAAGTCGACCGA ACCTCCGCCTCC-3' |
| Sequence-based reagent | YeMP-Nluc-Fwe-1D4 Reverse-2 | This paper | | 5'- AAAACGCGTTTACG CAGGCGCGACTTGG-3' |
| Sequence-based reagent | YeMP-Nluc-Fwe[K29A/R33A]−1D4, YeMP-Nluc-Fwe[K95A/K100A/R105A/K146A/K147/R150A]−1D4, YeMP-Nluc-Fwe[K29A/R33A/K95A/K100A/R105A/K146A/K147A/R150A]−1D4 forward | This paper | | 5'- AAAACTAGTGTCGAC TCG TTTGCGGAAAAGATAACG-3' |
| Sequence-based reagent | YeMP-Nluc-Fwe[K29A/R33A]−1D4, YeMP-Nluc-Fwe[K95A/K100A/R105A/K146A/K147/R150A]−1D4, YeMP-Nluc-Fwe[K29A/R33A/K95A/K100A/R105A/K146A/K147A/R150A]−1D4 Reverse | This paper | | 5'- AAAACGCG TTTACGCAGGCGCG ACTTGGCTGGTCTCTGTTG TGGGGCGCC-3' |
| Software, algorithm | LSM Zen | Zeiss | https://www.zeiss.com/microscopy/us/products/microscope-software/zen-lite.html | |
| Software, algorithm | Image J | https://imagej.net/contributors | https://imagej.nih.gov/ij/ | |

*Continued on next page*

*Appendix 1—key resources table continued*

| Reagent type (species) or resource | Designation | Source or reference | Identifiers | Additional information |
|---|---|---|---|---|
| Software, algorithm | pClamp 10.6 | Moleculardevices | https://www.moleculardevices.com/products/cellular-imaging-systems/acquisition-and-analysis-software/metamorph-microscopy#gref | |
| Software, algorithm | MetaMorph | Moleculardevices | https://www.moleculardevices.com/products/cellular-imaging-systems/acquisition-and-analysis-software/metamorph-microscopy#gref | |
| Software, algorithm | GraphPad Prism 8.0 | Prism | https://www.graphpad.com/ | |

