## [Decision Letter]

**Acceptance summary:**

Your work nicely reveals new aspects of the function of Flower in bulk endocytosis by regulating the levels of PI(4,5)P2. The observation that Flower recruits PI(4,5)P2 which directly enhances Flower channel activity is elegantly demonstrated. Moreover, the fact that PI(4,5)P2 drives ADBE which in turn removes Flower, nicely explains how the process is terminated.

**Decision letter after peer review:**

Thank you for submitting your article "Flower initiates a positive feedback loop upon PIP_2_ enrichment at periactive zones to control bulk endocytosis" for consideration by *eLife*. Your article has been reviewed by three peer reviewers, and the evaluation has been overseen by Hugo Bellen, Reviewing Editor and Richard Aldrich, Senior Editor. The following individual involved in review of your submission has agreed to reveal their identity: Patrik Verstreken (Reviewer #1).

The reviewers have discussed the reviews with one another and the Reviewing Editor has drafted this decision to help you prepare a revised submission. As you will notice the reviewers find your data interesting and they have a positive tone. However, they raise a number of issues. I have read their reviews and your manuscript. Many of the issues that are raised relate to better controls, textual changes, clarifications, and quantification of data. We think you can address most of these in a timely manner.

Reviewer #1:

In this paper Li et al. show how the calcium channel and PI(4,5)P2 regulate bulk endocytosis at synapses. Their work brings exciting new mechanistic insight into the regulation of this process, and what I find of particular interest is the notion that Fwe might be gated by PI(4,5)P2 to control the retrieval of synaptic membrane. I think this is an exciting paper and I have mostly minor comments on interpretation and one suggestion for a set of genetic interactions to cement the role for calcineurin in this pathway.

1) The authors elegantly confirm and demonstrate further that PI(4,5)P2levels increase during stimulation. They conclude this is in microdomains based on several pieces of data. This indeed seems to be the case but I think it is important to note that chemical fixation may alter distribution. I would either note this in the text or try to include single particle tracking experiments (eg sptPALM).

2) In Figure 2 they investigate the role for PI(4,5)P2 in ADBE using PLCdPH over expression and Synj over expression. It might be interesting to look at the role for Tweek in this process as well, giving the opposing effect of Tweek in comparison to Synj1.

3) I wonder if in the reformation assays in Figure 2 the authors can observe the formation of small SVs at the surface of bulk endosomes?

4) I think the conclusion that the PLA signal indicates a physical interaction between Fwe and PI(4,5)P2 is overstated: to conclude that Fwe moves in to PI(4,5)P2-enriched domains would be more correct. The demonstration of a direct interaction is given later in the paper in Figure 4.

5) The authors write "Similar to the effect of loss of Fwe, reduced CanA1 levels greatly suppressed the formation of PIP2 microdomains compared to the control (Figure 3E-F), indicating that Calcineurin mediates the Fwe Ca^2+^ channel to induce PIP2 microdomains." This is not shown: the authors show that both Fwe and CanA1 reduce the formation of PI(4,5)P2 microdomains during stimulation, but not that CanA1 mediates this effect in an Fwe-dependent manner. This should be shown using genetic interactions: eg test if double mutants show a similar (non-additive) effect compared to either single mutant, or overexpress Fwe in CanA1 mutants (and vice versa) to determine if this rescues the defect.

6) Figure 6G-H seem superfluous to me and not central to the story. I would remove this from the manuscript.

7) I also suggest to write PI(4,5)P2 or PhIns(4,5)P2 instead of PIP2 because the latter refers to all phosphoinositides that have 2 phosphate groups, and not specifically to PI(4,5)P2.

Reviewer #2:

This paper describes an intriguing model for a feedback pathway by which the calcium channel Flower, Ca++ signaling, and PIP2 together regulate activity dependent bulk endocytosis and synaptic vesicle regeneration at a neuronal synapse. This is an interesting and novel idea, which could illuminate key molecular dynamics downstream of synaptic activity that regulate synaptic vesicle recycling. The strengths of the paper are the in vitro-in vivo analysis of fwe-PIP2 binding and function, and in the measurement of each of PIP2, Calcium signaling, and membrane traffic for each component.

1) The authors claim the increase in fluorescence of PHPLC-GFP as evidence of clustering in microdomains. However, this is a circular argument, as the increase in fluorescence is proposed as the evidence that PIP2 microdomains have formed. This claim is an important premise of the paper, as PHPLC fluorescence intensity is the primary quantified metric for a central conclusion of the paper (that PIP2 microdomains form upon stimulation, and require fwe).

At a technical level, it is difficult to understand why clustering should increase fluorescence, since the total amount of protein is not changed by Western blotting. One possibility is that diffuse fluorescence is below the detection limit of the microscope, while clustered fluorescence is above this limit; however, comparing measurements in this way from inside and outside the linear range of detection is misleading. Another possibility is that GFP fluorescence emission (but not levels) is increased in the microenvironment of the puncta, but this is not evidence for clustering, but instead for a changed microenvironment. The authors could quantify this by post-staining after fixation with anti-GFP antibodies in a second channel (they would expect to see no clustering if the effect was purely on GFP fluorescence). Either way, the whole-bouton quantification presented here is not sufficient, and more rigorous quantification of clustering is required (see point 4).

Finally, there are concerns about potential effects of PH-PLCdelta (which has been used here and elsewhere both as a reporter and manipulation for PIP2). What level of expression were used in Figure 1 compared to 2, and could the "clusters" coincide with malfunctioning endocytosis? Levels should be compared directly (rather than inferred from temperature).

Due to these questions, some form of independent evidence (published or novel) for PIP2 clustering must be shown or cited.

2) The consistency of phenotypes among experimental manipulations on Calcium, PIP2, and vesicle dynamics is compelling evidence for a feedback loop. However, not all functional links in the loop are completely tested; for instance, whether CanA1 loss of function leads to calcium signaling defects via synj/PIP2, or through its myriad other targets. While it might be possible to “normalize” synj phosphorylation using minibrain heterozygotes or mutants, this is beyond the scope of suggested revisions. However, where a link in the chain has not been as rigorously tested (as for CanA1), the authors should discuss other possible interpretations of these data (as they do for other possible means of gating Flower).

3) Figure 2—figure supplement 2- why does FM get taken up to similar extent with high PLC or synj if EM data suggests ABDE is not happening? And why is unloading not impaired if vesicle recycling is diminished?

4) There are several instances where images are not quantified, or where the method of quantification is insufficiently described:

a) Currently Materials and methods indicate only that “image processing and quantification were achieved using LSM Zen and ImageJ”. The authors must describe in detail how fluorescence imaging was processed and quantified. What processing steps were performed? In 2D or 3D? Projections or slices? How were NMJ masks or ROIs created? Materials and methods do not describe what kind of microscope and objective were used for live imaging (important for understanding resolution). All this information is required to correctly interpret the imaging data.

b) Microdomains are poorly described: how is their size quantified? 200nm is very close to the resolution limit of SIM, suggesting many measurements may have been at the floor of the resolution of the system and that this measurement likely represents an overestimate of the size of the domain. It would be informative to see a histogram of values to determine to what degree this is the case, and discuss if appropriate.

c) How was DAB staining (Figure 6) quantified? What was the criterion used to discriminate between no labeling and “similar to SV” (eg in d, where the overall staining appears to be more similar to a than to b). In the text, the data are presented as both SV and bulk endosome levels decreasing, though these data are not quantified or presented in the figure.

5) If fwe initiates a positive feedback loop, why do only high levels of activity stimulate clustering and ADBE?

6) The claim that Fwe is “selectively sorted” by PIP2 seems overstated, especially given the companion idea that Fwe sorts PIP2 into microdomains. In general, the data in Figure 6 are a bit unconvincing or overinterpreted:

a) AP2 complexes strongly associated with PIP2 microdomains- this is completely unsupported by the data, which show an exceedingly low correlation.

b) How were “colocalized” pixels determined in Figure 6H (not described in legend or Materials and methods)? Further, the figure legend for 6I indicates that “Quantification data for Pearson correlation coefficients” is the overlapping area divided by the overall area. This is not a Pearson's correlation coefficient, which rather depends on the linear relationship between channel intensities of all (or a subset) of pixels.

c) “synj altered the localization of the AP2 complex”- this is not quantified and is very difficult to determine from the images presented. This is also a clear example where it is not obvious that images are being taken from the same Z-plane, which further confounds interpretation of the data.

d) AP2-fwe IPs are not very convincing. The difference between negative control and IP is subtle, only a very small fraction of input, and neither repeated or quantified. These data should be strengthened or removed.

e) Is synj expressed at room temperature or 29 degrees?

Reviewer #3:

Previous work by the Yao lab showed that the SV-associated Ca^2+^ channel Flower (Fwe) is required for both clathrin-mediated endocytosis (CME) of synaptic vesicles (SVs) from the plasma membrane (PM) and activity-dependent bulk endocytosis (ADBE). Importantly, Fwe has likely a distinct role for each endocytotic route since only ADBE requires Ca^2+^ channel function. This follow-up study pursues the mechanisms underlying Fwe's role for ABDE. The study provides evidence for an interesting model of Fwe's role in ADBE that may significantly advance our understanding of mechanisms underlying ADBE. Specifically, it suggests that freshly exocytosed Fwe on the PM forms a positive Ca^2+^-dependent feedback loop that facilitates the generation and accumulation of phosphatidylinositol 4,5-bisphosphate (PIP2) in microdomains on the PM. In turn, accumulating PIP2 may facilitate compartmentalization of Fwe to bulk endosomes, thereby terminating ADBE. I have a number of concerns.

1) To visualize PIP2 at fly NMJs, the entire study relies only on a single PIP2 marker, the Gal4-driven expression of PLCδ1-PH-EGFP. Unfortunately, this marker not only reduces "free" PIP2 levels but also has dominant-negative effects on synaptic growth and SV cycling (Khuong et al., 2010). To properly support the notion that intense stimulation causes accumulation of PIP2 in microdomains on the PM, a number of corrections are required:

a) the PLCδ1-PH-EGFP imaging experiments require a proper control, like the expression of PLCδ1-PHS39R-GFP, which does not bind PIP2. Measuring mCD-EGFP expression levels is not a proper control.

b) the stimulation-induced accumulation of PIP2 should be verified by a second independent marker like quantitative immunostainings for edogenous α-adaptin or other PIP2-binding proteins. This is especially important to rule out unaccounted dominant-negative effects.

c) It is unclear how changes in PLCδ1-PH-EGFP fluorescence (Figure 1A-B, D, plus all subsequent experiments using PLCδ1-PH-EGFP including Figure 3C-F, 4F-G) are measured at larval NMJs since no information about respective ROIs is provided. To make a strong statement about a PIP2 accumulation on the PM, PLCδ1-PH-EGFP fluorescence measured for an entire bouton (from one or a few slices) should be subtracted by the PLCδ1-PH-EGFP fluorescence obtained from the cytosol of the bouton. Measurements using ROIs for the entire bouton do not allow specific conclusions about PM levels.

2) To significantly strengthen the conclusion that PIP2 accumulation drives ADBE initiation and SV reformation (Figure 2) a proper control should be used for the postulated PLCδ1-PH-EGFP-mediating reduction of PIP2 levels. Ideally, one may use expression of the PLCδ1-PHS39R-GFP mutant. Overexpression of synaptojanin alone does not make the point that the reduced ADBE levels are due to PIP2 depletion since synaptojanin also affects several other phosphatidylinositides.

3) The proximity ligation experiments of Figure 3A require a proper control like PLCδ1-PHS39R-GFP co-expression with Flag-Fwe-HA. Otherwise, the observed proximity could be due to other factors.

4) The images of Figure 3C, e show little evidence of PLCδ1-PH-EGFP microdomains at the PM, much in contrast to Figure 1. Does this mean that the quantification associated with these figures mostly reflects PIP2 on cytosolic bulk endosomes?

5) The shown knockdown data of CanA1 in Figure 3E-H require a control for specificity, best co-expression of an RNAi-resistant WT transgene.

6) The study suggests that PIP2 binding to freshly exocytosed Fwe gates Fwe's Ca^2+^ channel activity. In turn, the resulting Ca^2+^ influx activates calcineurin and the postulated feedback loop accumulating PIP2 at the PM. If correct, one would expect that a mutation that several affects Fwe' Ca^2+^ channel activity (like FweE79Q) should significantly reduce the accumulation of PIP2. Is this indeed the case? If not, the Fwe-dependent PIP2 accumulation may occur through a different mechanism.

7) The HA-Fwe-APEX2 experiments of Figure 6 require further controls including documentation that the fusion protein is properly localized to SVs and not excessively retained on the PM during ADBE. The EM alone is not convincing.

8) The co-immunoprecipitation of Fwe with AP2 shown in Figure 6G requires quantification.

9) The evidence for the conclusion that accumulating PIP2 facilitates "sorting" of Fwe for ADBE is weak because it essentially rests on the HA-Fwe-APEX2 experiment. Since reducing PIP2 levels by synaptojanin expression reduced the bulk endosome localization of Fwe as judged by DAB staining levels (Figure 6D-F), the authors conclude that "PIP2 microdomains can facilitate the retrieval of Fwe to the bulk endosome". However, this appears to be somewhat a catch22 since reduced PIP2 levels decrease the overall amount of bulk endocytosis. Whether the potentially reduced levels of internalized Fwe are disproportionally reduced has not been addressed. In addition, DAB staining intensity should not be used to quantify protein levels as the enzymatic reaction may have different constrains on SVs versus endosomes.

[Editors' note: further revisions were suggested prior to acceptance, as described below.]

Thank you for resubmitting your work entitled "A positive feedback loop between Flower and PI(4,5)P_2_ at periactive zones controls bulk endocytosis in *Drosophila*" for further consideration by *eLife*. Your revised article has been evaluated by Richard Aldrich (Senior Editor) and Hugo Bellen (Reviewing Editor).

The manuscript has been improved but there are some remaining issues that need to be addressed before acceptance, as outlined below:

The authors have provided a meaningfully improved manuscript that largely addresses the major concerns of the paper. In particular, the authors provide several new lines of evidence that levels or organization of PI(4,5)P2 are likely changing in response to stimulation: There is no change in the new negative control PH S39 mutant, AP2a levels increase at the membrane, and the levels of native and anti-GFP fluorescence of PH-PLC increase at the membrane. Especially given the strong correspondence between in vitro and in vivo data and the consistent phenotypes among the perturbations in the proposed pathway, these new data further strengthen the main claim that Fwe acts in a positive feedback loop with Ca++ and PI(4,5)P2.

It is worth noting that the precise nature of the microdomains remains a bit unclear. It is difficult to discern whether PI(4,5)P2 microdomains are forming de novo following stimulation, or whether PI(4,5)P2 ¬signal increases in existing microdomains (for example, AP2a recruitment to the membrane is quite high even under resting conditions). And as noted by the authors, while the PH domain is the best available tool to answer the questions posed, it remains possible that properties of the microdomains partially reflect the function-perturbing aspects of the probe. However, these caveats do not significantly detract from the core conclusions of the paper.

Figure 3H. Something is wrong with the shown KD data by UAS-canA1-RNAi(FB5) or the drawn conclusion that "reducing CanA1 levels via expression of either RNAi construct greatly suppressed the formation of PI(4,5)P2 microdomains compared to the control (Figure 3G-H)". The resting levels for the canA1 KD are only 50% of control. The K^+^-induced increase PLCδ1-PH-EGFP in the KD is as large as in control. Accordingly, the KD seems to have no effect, which would suggest that one of the two KDs is not specific for canA1.

If the FB5 KD is not working, the one needs a wild type CanA1 co-expression for the KD with the canA1-RNAi(JF01871) transgene to demonstrate specificity.

Please clarify.

Figure 3I. The canA1 KD uses the controversial UAS-canA1-RNAi(FB5) transgene, which does not appear to block the formation of PI(4,5)P2 microdomains (Figure 3H). The experiment should be done with the UAS-canA1-RNAi(JF01871) transgene as only this transgene reliable affects PI(4,5)P2 microdomains (assuming the KD can be rescued). Otherwise, one cannot conclude that "Calcineurin mediates the Ca^2+^ influx conducted by Fwe to induce PI(4,5)P2 microdomains". Resolving this issue is critical.

"Together, these data suggest that the PI(4,5)P2 microdomains initiate ADBE to control SV membrane retrieval upon intense stimulation." This sentence should be rephrased. The shown FM1-43 data (Figure 2—figure supplement 2B) only suggest that overexpression of PLCδ1-PH-EGFP or Synj impairs SV endocytosis independent of the mode of endocytosis. The shown FM1-43 data do not exclude a role of PI(4,5)P2 for other modes of endocytosis.

Moreover, the FM1-43 data use a much shorter (30 seconds) and milder stimulation than the solid EM data visualizing ADBE (10 minutes; Figure 2). Therefore, the FM1-43 and EM data are not directly comparable.

The same concern applies to the FM1-43-visulaized SV exocytosis data, which indicate that SV exocytosis is not impaired by overexpression of PLCδ1-PH-EGFP or Synj if endocytosis is stimulated for 1 minute. Again, the data are not comparable to the ADBE defect shown in Figure 2. I suggest either to omit the data of Figure 2—figure supplement 2 or use the same stimulation paradigm for the FM1-43 and EM data.

---

## [Author Response]

Reviewer #1:In this paper Li et al. show how the calcium channel and PI(4,5)P2 regulate bulk endocytosis at synapses. Their work brings exciting new mechanistic insight into the regulation of this process, and what I find of particular interest is the notion that Fwe might be gated by PI(4,5)P2 to control the retrieval of synaptic membrane. I think this is an exciting paper and I have mostly minor comments on interpretation and one suggestion for a set of genetic interactions to cement the role for calcineurin in this pathway.1) The authors elegantly confirm and demonstrate further that PI(4,5)P2levels increase during stimulation. They conclude this is in microdomains based on several pieces of data. This indeed seems to be the case but I think it is important to note that chemical fixation may alter distribution. I would either note this in the text or try to include single particle tracking experiments (eg sptPALM).

For our live-imaging experiments (also see Materials and methods), we stimulated the NMJs with 40-Hz electric stimulation or a 90-mM K^+^ chemical stimulation under conditions of 2 mM extracellular Ca^2+^. However, these stimuli caused pronounced muscle contraction, which prevented us from precisely focusing the objective lens on a single NMJ bouton over the imaging timespan for confocal microscopy. Attempts to slow down contractions by desensitizing the glutamate receptors with 7 mM glutamate treatment were unsuccessful. Therefore, in our experiments, we manually adjusted the focus plane and captured snapshot images of the same single boutons each second for one minute during stimulation by using a long working distance water immersion objective (XLUMPLFLN20XW water lens) and an EMCCD camera (iXon, Andor) mounted on a SliceScope Pro 6000 (Scientifica) microscope. For each imaging experiment, at least three focused images of the same boutons under resting, stimulation, or post-stimulation conditions were used for data quantification. Fluorescence intensities of PLC_δ1_-PH-EGFP within the boutons were assessed using ImageJ and averaged to serve as one individual data point. Due to these technical issues, it is difficult to perform single-particle tracking.

We have assessed the possibility that chemical fixation may alter membrane properties and thereby account for plasma membrane clustering of PLC_δ1_-PH-EGFP upon intense stimulation. We have now examined the distribution of mCD8-GFP and PLC_δ1_-PHS39R-EGFP (as also suggested by reviewer #3), a PI(4,5)P_2_-binding mutant Khuong et al., 2010Verstreken et al., 2009(; ). We found that there was no obvious change in the plasma membrane pattern of mCD8-GFP in fixed boutons stimulated with high K^+^ (Figure 1—figure supplement 2A-B). Unlike PLC_δ1_-PH-EGFP, PLC_δ1_-PHS39R-EGFP was observed mainly in the cytosol at rest. After intense stimulation, immunostaining signals of PLC_δ1_-PHS39R-EGFP did not accumulate on the plasma membrane (Figure 1—figure supplement 2C-D) and did not increase within the boutons (Figure 1—figure supplement 2E). In addition, we have further assessed potential dominant-negative effects of PI(4,5)P_2_ binding by PLC_δ1_-PH-EGFP on the stimulation-induced accumulation of PI(4,5)P_2_ by examining recruitment of the AP-2 complex by PI(4,5)P_2_. Similar to PLC_δ1_-PH-EGFP, levels of the α subunit of the AP-2 complex (AP-2α) were also increased on the plasma membrane upon high K^+^ stimulation (Figure 1—figure supplement 3A-B). Taken together, these results suggest that PLC_δ1_-PH-EGFP clustering on the plasma membrane upon stimulation arises from an increase in local PI(4,5)P_2_ concentration, although a potential effect of chemical fixation, if any, on PLC_δ1_-PH-EGFP clustering cannot be excluded. The new data from PLC_δ1_-PHS39R-EGFP and AP-2α experiments have been added to the figures indicated in our explanation here, and a respective statement has been added to the main text.

2) In Figure 2 they investigate the role for PI(4,5)P2 in ADBE using PLCdPH over expression and Synj over expression. It might be interesting to look at the role for Tweek in this process as well, giving the opposing effect of Tweek in comparison to Synj1.

We agree it would be interesting to test for a potential role of Tweek in ADBE, although we also feel it may be beyond the scope of this paper. We did attempt to obtain *tweek* mutant stocks (*tweek^1^* and *tweek^2^*) from the Bloomington *Drosophila* Stock Center and other sources. Unfortunately, after multiple requests, due to the hot summer weather, the mutant stocks died during delivery. Therefore, we could not perform the suggested TEM analysis within the limited revision period.

3) I wonder if in the reformation assays in Figure 2 the authors can observe the formation of small SVs at the surface of bulk endosomes?

We thank the reviewer for raising this interesting point. We have carefully re-examined our TEM images collected during recovery periods. However, we did not observe any accumulation of SVs budding from the bulk endosome in control boutons or boutons in which the PI(4,5)P_2_ microdomains were perturbed. This observation further indicates that PI(4,5)P_2_ microdomain formation likely plays an early role in promoting SV reformation from the bulk endosome. We now state this observation in the revised manuscript.

4) I think the conclusion that the PLA signal indicates a physical interaction between Fwe and PI(4,5)P2 is overstated: to conclude that Fwe moves in to PI(4,5)P2-enriched domains would be more correct. The demonstration of a direct interaction is given later in the paper in Figure 4.

As suggested by this reviewer and also by reviewer #3, we have now rephrased our conclusion to: “The PLA signal was low in resting boutons, whereas high K^+^ treatment significantly increased PLA signal intensity (Figure 3A-B). Therefore, these results suggest that Fwe and PI(4,5)P_2_ are closely colocalized when intense stimulation triggers PI(4,5)P_2_ microdomain formation.”

5) The authors write "Similar to the effect of loss of Fwe, reduced CanA1 levels greatly suppressed the formation of PIP2 microdomains compared to the control (Figure 3E-F), indicating that Calcineurin mediates the Fwe Ca^2+^ channel to induce PIP2 microdomains." This is not shown: the authors show that both Fwe and CanA1 reduce the formation of PI(4,5)P2 microdomains during stimulation, but not that CanA1 mediates this effect in an Fwe-dependent manner. This should be shown using genetic interactions: eg test if double mutants show a similar (non-additive) effect compared to either single mutant, or overexpress Fwe in CanA1 mutants (and vice versa) to determine if this rescues the defect.

As suggested, we have now assessed if Fwe overexpression can rescue the ADBE defect caused by *canA1* RNAi knockdown. Neuronal *canA1* was knocked down by expressing two independent *canA1* RNAi lines, which impaired ADBE relative to the *nsyb-GAL4* control (Figure 5A-B)[To address the similar concern raised by reviewer #3, we have conducted an additional *canA1* RNAi knockdown experiment using *UAS-canA1-RNAi* (JF01871)]. We found that Fwe overexpression was able to significantly reverse the ADBE defect under both of these *canA1* RNAi knockdown conditions (Figure 5A-B). Therefore, increasing Fwe-dependent Ca^2+^ influx can augment activation of the residual CanA1 enzymes and thus normalize downstream ADBE. These new data support that CanA1 mediates this effect in an Fwe-dependent manner. The new data have been added to Figure 5A-B, and a respective statement has been added to the main text.

We have also attempted to express a constitutive form of CanA1 in the *fwe* mutant background using two different neuronal *GAL4* drivers, *nsyb-GAL4* and *elav-GAL4.* Whereas single expression of a constitutive form of CanA1 did not affect larval viability, this expression caused precocious lethality of *fwe* mutants at very early larval stages. Unfortunately, the tiny size of the resulting larvae prevented us from further NMJ characterization.

6) Figure 6G-H seem superfluous to me and not central to the story. I would remove this from the manuscript.

We thank the reviewer for this constructive suggestion. We have removed these data from Figure 6.

7) I also suggest to write PI(4,5)P2 or PhIns(4,5)P2 instead of PIP2 because the latter refers to all phosphoinositides that have 2 phosphate groups, and not specifically to PI(4,5)P2.

Acknowledged. We have now changed PIP_2_ to PI(4,5)P_2_ in the revised manuscript.

Reviewer #2:This paper describes an intriguing model for a feedback pathway by which the calcium channel Flower, Ca++ signaling, and PIP2 together regulate activity dependent bulk endocytosis and synaptic vesicle regeneration at a neuronal synapse. This is an interesting and novel idea, which could illuminate key molecular dynamics downstream of synaptic activity that regulate synaptic vesicle recycling. The strengths of the paper are the in vitro-in vivo analysis of fwe-PIP2 binding and function, and in the measurement of each of PIP2, Calcium signaling, and membrane traffic for each component.1) The authors claim the increase in fluorescence of PHPLC-GFP as evidence of clustering in microdomains. However, this is a circular argument, as the increase in fluorescence is proposed as the evidence that PIP2 microdomains have formed. This claim is an important premise of the paper, as PHPLC fluorescence intensity is the primary quantified metric for a central conclusion of the paper (that PIP2 microdomains form upon stimulation, and require fwe).At a technical level, it is difficult to understand why clustering should increase fluorescence, since the total amount of protein is not changed by Western blotting. One possibility is that diffuse fluorescence is below the detection limit of the microscope, while clustered fluorescence is above this limit; however, comparing measurements in this way from inside and outside the linear range of detection is misleading. Another possibility is that GFP fluorescence emission (but not levels) is increased in the microenvironment of the puncta, but this is not evidence for clustering, but instead for a changed microenvironment. The authors could quantify this by post-staining after fixation with anti-GFP antibodies in a second channel (they would expect to see no clustering if the effect was purely on GFP fluorescence).

We thank the reviewer for raising this concern. We have now performed the suggested experiments. Our confocal images revealed that, for neurons at rest, there was weak fluorescence signal of native PLC_δ1_-PH-EGFP on the presynaptic plasma membrane labelled by α-Hrp staining, with some additional signal dispersed in the cytosol (Figure 1—figure supplement 1C). First, we elevated PI(4,5)P_2_ levels on the plasma membrane by removing a copy of *synaptojanin* (*synj*), which encodes the major neuronal PI(4,5)P_2_ phosphatase (Tsujishita et al., 2001; Verstreken et al., 2003). The resulting reduced level of Synj enhanced PLC_δ1_-PH-EGFP fluorescence (green channel, Figure 1—figure supplement 1C-D), consistent with previous studies (Chen et al., 2014; Verstreken et al., 2009). In this context, we did not observe a significant change in protein expression of PLC_δ1_-PH-EGFP in the presynaptic compartment (Figure 1—figure supplement 1E-F). By using α-GFP immunostaining (red channel), these PLC_δ1_-PH-EGFP signals could be faithfully amplified (Figure 1—figure supplement 1C-D).

Next, we stimulated the boutons expressing PLC_δ1_-PH-EGFP and treated with 90 mM K^+^ and 2 mM Ca^2+^ for 10 min. Similar to our live-imaging results, both native fluorescence and α-GFP immunostaining signals of PLC_δ1_-PH-EGFP were significantly increased on the presynaptic plasma membrane (Figure 1—figure supplement 1G-H) In particular, we observed high-level induction of PLC_δ1_-PH-EGFP puncta (Figure 1C-D). As a control experiment, we examined the distribution of mCD8-GFP and PLC_δ1_-PHS39R-EGFP, a PI(4,5)P_2_-binding mutant (Khuong et al., 2010; Verstreken et al., 2009) (as suggested by reviewer #3), which revealed no obvious change in the plasma membrane pattern of mCD8-GFP in fixed boutons stimulated with high K^+^ (Figure 1—figure supplement 2A-B). Unlike PLC_δ1_-PH-EGFP, PLC_δ1_-PHS39R-EGFP was found mainly in the cytosol at rest. After intense stimulation, immunostaining signals of PLC_δ1_-PHS39R-EGFP did not accumulate on the plasma membrane (Figure 1—figure supplement 2C-D) and were not increased (Figure 1—figure supplement 2E). As suggested by reviewer #3, we have now assessed recruitment of the AP-2 complex upon PI(4,5)P_2_ increase. Similar to PLC_δ1_-PH-EGFP, levels of the α subunit of the AP-2 complex (AP-2α) were also increased on the plasma membrane upon high K^+^ stimulation (Figure 1—figure supplement 3A-B). Thus, the clustering of PLC_δ1_-PH-EGFP on the plasma membrane upon stimulation results from an increase in local PI(4,5)P_2_ concentration. As pointed out by this reviewer, we think that the signal of the diffuse PLC_δ1_-PH-EGFP probe in *Drosophila* NMJ boutons is below the detection limit of confocal microscopy, whereas the signal from the clustered probe on the plasma membrane exceeds this limit and thus is much more detectable.

These new data have been added to the above-mentioned figures, and a respective statement has been added to the revised manuscript.

Either way, the whole-bouton quantification presented here is not sufficient, and more rigorous quantification of clustering is required (see point 4).

Since the signals of PLC_δ1_-PH-EGFP are mainly associated with the plasma membrane in boutons (Figure 1—figure supplement 1), we previously quantified the immunostaining signals of PLC_δ1_-PH-EGFP from the entire bouton outlined by α-Hrp staining to indicate any change in the plasma membrane association of PLC_δ1_-PH-EGFP. As suggested, we have now re-quantified PLC_δ1_-PH-EGFP signal on the presynaptic plasma membrane defined by α-Hrp staining.

For details, please also see our response to point 4.

Finally, there are concerns about potential effects of PH-PLCdelta (which has been used here and elsewhere both as a reporter and manipulation for PIP2). What level of expression were used in Figure 1 compared to 2, and could the "clusters" coincide with malfunctioning endocytosis? Levels should be compared directly (rather than inferred from temperature).

We apologize for the confusion. In Figure 1, we expressed PLC_δ1_-PH-EGFP using *nsyb-GAL4* at 25 °C. We also quantified the level of PLC_δ1_-PH-EGFP expressed at 25 °C vs 29 °C. As shown in Figure 2—figure supplement 1A-B, compared to its expression at 25 °C, elevating the temperature to 29 °C led to a two-fold increase in probe expression. This information has now been added to the revised manuscript. We also now state the experimental temperatures in all figure legends.

It is conceivable that PLC_δ1_-PH-EGFP can suppress the binding of PI(4,5)P_2_ effectors and PI(4,5)P_2_ catalytic enzymes to PI(4,5)P_2_ when this reporter is highly expressed. For all figures, we measured the level of PI(4,5)P_2_ under conditions of mild PLC_δ1_-PH-EGFP expression (reared at 25 °C). This expression level only mildly affects normal ADBE (Figure 2C-D). Therefore, such a mild ADBE defect should not account for PI(4,5)P_2_ clustering upon intense stimulation. In support of this notion, similar to PLC_δ1_-PH-EGFP, the AP-2 complex was recruited to the plasma membrane upon high K^+^ stimulation (Figure 1—figure supplement 3A-C). However, we cannot exclude the possibility that PLC_δ1_-PH-EGFP expression may suppress the binding of PI(4,5)P_2_ catalytic enzymes to slow rapid PI(4,5)P_2_ turnover, thereby enabling us to capture PI(4,5)P_2_ clustering. Nevertheless, our data suggest that PI(4,5)P_2_ microdomains are formed upon intense stimulation.

Due to these questions, some form of independent evidence (published or novel) for PIP2 clustering must be shown or cited.

As also suggested by reviewer #3, we have now investigated recruitment of the AP-2 complex upon PI(4,5)P_2_ increase. Similar to PLC_δ1_-PH-EGFP, levels of the α subunit of the AP-2 complex (AP-2α) were also increased and clustered on the plasma membrane upon high K^+^ stimulation (Figure 1—figure supplement 3A-B). This finding represents further evidence of PI(4,5)P_2_ clustering on the plasma membrane upon stimulation.

Moreover, we now cite additional studies showing PI(4,5)P_2_ clustering in different cellular contexts.

2) The consistency of phenotypes among experimental manipulations on Calcium, PIP2, and vesicle dynamics is compelling evidence for a feedback loop. However, not all functional links in the loop are completely tested; for instance, whether CanA1 loss of function leads to calcium signaling defects via synj/PIP2, or through its myriad other targets. While it might be possible to “normalize” synj phosphorylation using minibrain heterozygotes or mutants, this is beyond the scope of suggested revisions. However, where a link in the chain has not been as rigorously tested (as for CanA1), the authors should discuss other possible interpretations of these data (as they do for other possible means of gating Flower).

We have now added further discussion on this topic in the revised manuscript.

3) Figure 2—figure supplement 2- why does FM get taken up to similar extent with high PLC or synj if EM data suggests ABDE is not happening? And why is unloading not impaired if vesicle recycling is diminished?

For our loading/unloading FM4-64 dye uptake assay, we stimulated the NMJ boutons with high K^+^ to take up the dye for a long period of time, e.g. 5 minutes, so that a majority of both the exo-endo cycling pool (ECP) and reserved pool (RP) were loaded with FM dye. Then we stimulated them with high K^+^ for 1 minute to release the loaded dye. By calculating the ratio of unloaded dye to loaded dye, we could rule out the effect of SV endocytosis and measure the efficacy of SV exocytosis. Our results revealed no effect on the rate of SV exocytosis when the PLC_δ1_-PH domain or Synj was expressed. Since we did not image the fluorescence intensities of FM dye under the same imaging detection conditions, the signal intensities of loaded dye by endocytosis cannot be compared directly.

To address the reviewer’s question, we have conducted a FM1-43 dye uptake assay. We stimulated the boutons with a 30-second pulse under conditions of 90 mM K^+^ and 2 mM Ca^2+^ to trigger SV exocytosis and prompt the recycling SVs to take up FM dye (Figure 2—figure supplement 2A). We imaged the boutons under different genotypes with same imaging setup. Whereas control boutons exhibited appropriate dye uptake, overexpression of either PLC_δ1_-PH-EGFP or Synj impaired dye uptake (Figure 2—figure supplement 2B-C). These findings are consistent with ADBE being defective. This new data has been added to Figure 2—figure supplement 2A-C and a respective statement has been added to the revised manuscript.

4) There are several instances where images are not quantified, or where the method of quantification is insufficiently described:a) Currently Materials and methods indicate only that “image processing and quantification were achieved using LSM Zen and ImageJ”. The authors must describe in detail how fluorescence imaging was processed and quantified. What processing steps were performed? In 2D or 3D? Projections or slices? How were NMJ masks or ROIs created? Materials and methods do not describe what kind of microscope and objective were used for live imaging (important for understanding resolution). All this information is required to correctly interpret the imaging data.

We apologize for this oversight. We have now ensured that quantification data now accompanies all images. Moreover, detailed procedures for our imaging experiments are now provided in the revised Materials and methods (as follows):

For live-imaging of PLC_δ1_-PH-EGFP:

“Ten images were taken from larval fillets at rest. Muscle contraction significantly decelerated after stimulation for 2 min. Thus, we captured 60 consecutive snapshot images every second from the third minute. Muscles 6 and 7 of abdominal segment 3 were imaged. Under the condition of high K^+^ stimulation, larval fillets were bathed in a solution of 90 mM K^+^/2 mM Ca^2+^/7 mM glutamate (25 mM NaCl, 90 mM KCl, 10 mM MgCl2, 10 mM NaHCO_3_, 5 mM trehalose, 5 mM HEPES (pH 7.4), 30 mM sucrose, 2 mM CaCl_2_, 7 mM monosodium glutamate) for 5 min. Sixty consecutive snapshot images were captured every second from the fifth minute of stimulation. Images were taken using a long working distance water immersion objective (XLUMPLFLN20XW, Olympus) and EMCCD camera (iXon, Andor) mounted on a SliceScope Pro 6000 (Scientifica) microscope and employing MetaFluor software (Molecular Devices). For each imaging experiment, at least three focused images for the same boutons under resting, stimulation, or post-stimulation conditions were used for data quantification. Fluorescence intensities of PLC_δ1_-PH-EGFP within the boutons were quantified using ImageJ and averaged to serve as one individual data-point. Image processing was achieved using LSM Zen.”

For live-imaging of GCaMP6f :

“Ten images were taken from larval fillets at rest. Subsequently, larval fillets were stimulated with a solution of 90 mM K^+^/2 mM Ca^2+^/7 mM glutamate for 5 min. Sixty consecutive snapshot images were captured every second from the fifth minute of stimulation. The NMJs were derived from muscles 6 and 7 of abdominal segment 2/3. Images were taken using a water immersion objective (W Plan-Apochromat 40x/1.0 DIC M27, Zeiss). For each imaging experiment, at least three focused images of the same boutons under resting, stimulation, or post-stimulation conditions were used for data quantification. Fluorescence intensities of GCaMP6f within the boutons were quantified using ImageJ and averaged to serve as one individual data-point. Image processing was achieved using LSM Zen.”

For PLC_δ1_-PH-EGFP or AP-2α immunostaining signal:

“Consecutive single-plane images of the boutons of muscles 6 and 7 in abdominal segments 2 or 3 of all different experimental sets were taken using a Zeiss LSM 780 confocal microscope with a 63X 1.6 numerical aperture oil-immersion objective under a 1-μm interval setup and equal laser power and laser exposure time. The zoom value was adjusted to 2 to obtain maximal imaging resolution. For data quantification, single-plane images of five different individual boutons from each NMJ bouton image were used. The presynaptic plasma membrane regions of the type Ib boutons defined by α-HRP immunostaining were outlined manually, and native fluorescence or α-GFP immunostaining signal intensities of PLC_δ1_-PH-EGFP or AP-2α on the plasma membrane were quantified using ImageJ and averaged to serve as one individual data-point. Image processing was achieved using LSM Zen.”

For PLA:

“To quantitatively compare PLA signal, fixed larval fillets derived from different experimental conditions were collected into the same Eppendorf tube and processed. Consecutive single-plane images of the boutons of muscles 6 and 7 in abdominal segments 2 or 3 of all different experimental sets were taken using a Zeiss LSM 780 confocal microscope with a 63X 1.6 numerical aperture oil-immersion objective under a 1-μm interval setup and equal laser power and laser exposure time. The zoom value was adjusted to 2 to obtain maximal imaging resolution. For data quantification, consecutive single-plane images were projected, and the type Ib boutons were outlined based on the region of PLC_δ1_-PH-EGFP immunostaining. PLA or antibody immunostaining signal intensities were quantified using ImageJ and averaged to serve as one individual data-point. Image processing was achieved using LSM Zen.”

For FM1-43 dye assay:

“Consecutive bouton images of different fly genotypes were captured with a Zeiss 780 confocal microscope with a 63 x 1.6 numerical aperture oil-immersion objective under a 1-μm interval setup and equal laser power and laser exposure time. For data quantification, consecutive single-plane images were projected, and the type Ib boutons were outlined according to the region of FM dye labeling. FM dye signal intensities were quantified using ImageJ and averaged to serve as one individual data-point. Image processing was achieved using LSM Zen.”

For FM4-64 dye loading/unloading assay:

“Bouton snapshot images were captured using a water immersion objective (W Plan-Apochromat 40x/1.0 DIC M27, Zeiss) and EMCCD camera (iXon, Andor) mounted on a SliceScope Pro 6000 (Scientifica) microscope and employing MetaFluor software (Molecular Devices). FM dye fluorescence intensities in type Ib boutons from each image were quantified using ImageJ and averaged to serve as one individual data-point. Image processing was achieved using LSM Zen.”

If the reviewer still requests more details, we will add accordingly.

b) Microdomains are poorly described: how is their size quantified? 200nm is very close to the resolution limit of SIM, suggesting many measurements may have been at the floor of the resolution of the system and that this measurement likely represents an overestimate of the size of the domain. It would be informative to see a histogram of values to determine to what degree this is the case, and discuss if appropriate.

We thank the reviewer for raising this issue. To quantify SIM imaging data, we chose single-plane SIM images focused on the central section of individual boutons and outlined the immunostained and plasma membrane-associated PLC_δ1_-PH-EGFP clusters using ImageJ. The area of these clusters for each individual bouton was measured using ImageJ and then averaged to serve as one data-point. We assessed 37 boutons derived from five NMJs of three different larvae. The area of the clusters in each individual bouton was measured using Image J, and the areas of the clusters over 0.032 μm^2^ (equal to a ~200-nm diameter circle, i.e., the resolution limit of SIM) were averaged to serve as one data-point. We calculated an average cluster area of 0.07655 μm^2^ (equal to a ~300-nm diameter circle). We have now revised our statement in the main text. The requalified data have been added to Figure 1F, and respective details have been added to the Materials and methods.

c) How was DAB staining (Figure 6) quantified? What was the criterion used to discriminate between no labeling and “similar to SV” (eg in d, where the overall staining appears to be more similar to a than to b). In the text, the data are presented as both SV and bulk endosome levels decreasing, though these data are not quantified or presented in the figure.

To also ease the reviewer #3’s concern that “DAB staining intensity should not be used to quantify protein levels as the enzymatic reaction may have different constrains on SVs versus endosomes”, we have now re-quantified the DAB signal intensity ratio of bulk endosome to SV to indicate the protein levels of HA-Fwe-APEX2 on bulk endosomes, relative to those on the SVs. With this normalization, we can verify the role of PI(4,5)P_2_ microdomains in Fwe localization on bulk endosomes. We provide details of description and data quantification in the revised main text, Materials and methods, and figure legends as follows:

“To minimize staining variability from bouton to bouton, we compared DAB staining intensities on bulk endosomes and SVs from the same boutons. As shown in Figure 6B-F, compared to Flag-Fwe-HA rescue boutons, in HA-Fwe-APEX2 rescue boutons, the staining intensities of bulk endosomes were more abundant compared to those of the surrounding SVs, revealing a mechanism by which Fwe is recycled to bulk endosomes after it initiates ADBE.”

When we perturbed PI(4,5)P_2_ microdomains by Synj overexpression (Figure 2B), the bulk endosome localization of HA-Fwe-APEX2 was significantly reduced (Figure 6D-F), but there was only a mild reduction in SV localization (Figure 6D-E). Therefore, in addition to initiating ADBE, PI(4,5)P_2_ microdomains play a role in facilitating the retrieval of Fwe to the bulk endosome, enabling ADBE to remove its trigger via a negative feedback regulatory mechanism, thereby reducing endocytosis to prevent excess membrane uptake. We now provide this explanation in the revised manuscript

For data quantification of DAB intensities, the display color of TEM images was reverted to grayscale using ImageJ. Average DAB staining intensity for each individual bulk endosome was assessed. Then the average DAB staining intensities of 50-100 surrounding SVs from the same bouton image were used to assess the relative level of HA-Fwe-APEX2 on bulk endosomes vs SVs.

We add quantification data for the DAB staining intensity ratio of bulk endosomes to SVs the surrounding SVs in Figure 6F. The number of bulk endosomes, NMJ boutons, and larvae counted (Flag-Fwe-HA rescue control: Bulk endosomes (n=33) derived from 5 NMJ boutons of two different larvae; HA-Fwe-APEX2 rescue: Bulk endosomes (n=47) derived from 6 NMJ boutons of two different larvae; HA-Fwe-APEX2 rescue expressing Synj: Bulk endosomes (n=59) derived from 9 NMJ boutons of two different larvae. Individual values were shown in graphs. This description is now stated in figure legends.

5) If fwe initiates a positive feedback loop, why do only high levels of activity stimulate clustering and ADBE?

Our data show that direct binding of PI(4,5)P_2_ is required for the Ca^2+^ channel activity of Fwe. Perturbation of PI(4,5)P_2_-Fwe binding further impaired the formation of PI(4,5)P_2_ microdomains, as well as ADBE initiation. Hence, PI(4,5)P_2_ controls Fwe gating, so that Fwe can promote PI(4,5)P_2_ compartmentalization through positive feedback regulation. Furthermore, loss of Fwe impaired the intracellular Ca^2+^ increase evoked upon strong activity stimulation (Figure 3I-J) (Yao et al., 2017). These results indicate that, in addition to PI(4,5)P_2_, the channel function of Fwe may be gated by a significant change in membrane potential. Therefore, it is possible that both factors may gate Fwe, thereby only allowing channel opening when exocytosis targets Fwe to periactive zones. Future studies should explore the details of this channel gating mechanism. These points have now been added to our revised Discussion.

6) The claim that Fwe is “selectively sorted” by PIP2 seems overstated, especially given the companion idea that Fwe sorts PIP2 into microdomains. In general, the data in Figure 6 are a bit unconvincing or overinterpreted:

Acknowledged. We have now toned down this conclusion and revised the statement in our Introduction to: “PI(4,5)P_2_ also participates in retrieval of Fwe to the bulk endosome, thereby stopping membrane recycling.”. We have also revised our Abstract and Results.

a) AP2 complexes strongly associated with PIP2 microdomains- this is completely unsupported by the data, which show an exceedingly low correlation.b) How were “colocalized” pixels determined in Figure 6H (not described in legend or Materials and methods)? Further, the figure legend for 6I indicates that “Quantification data for Pearson correlation coefficients” is the overlapping area divided by the overall area. This is not a Pearson's correlation coefficient, which rather depends on the linear relationship between channel intensities of all (or a subset) of pixels.c) “synj altered the localization of the AP2 complex”- this is not quantified and is very difficult to determine from the images presented. This is also a clear example where it is not obvious that images are being taken from the same Z-plane, which further confounds interpretation of the data.d) AP2-fwe IPs are not very convincing. The difference between negative control and IP is subtle, only a very small fraction of input, and neither repeated or quantified. These data should be strengthened or removed.e) Is synj expressed at room temperature or 29 degrees?

As suggested by reviewer #1, since our data on Fwe and AP-2 complex association (original Figure 6G-H and text) were deemed superfluous, we have removed this topic from the revised manuscript.

Reviewer #3:Previous work by the Yao lab showed that the SV-associated Ca^2+^ channel Flower (Fwe) is required for both clathrin-mediated endocytosis (CME) of synaptic vesicles (SVs) from the plasma membrane (PM) and activity-dependent bulk endocytosis (ADBE). Importantly, Fwe has likely a distinct role for each endocytotic route since only ADBE requires Ca^2+^ channel function. This follow-up study pursues the mechanisms underlying Fwe's role for ABDE. The study provides evidence for an interesting model of Fwe's role in ADBE that may significantly advance our understanding of mechanisms underlying ADBE. Specifically, it suggests that freshly exocytosed Fwe on the PM forms a positive Ca^2+^-dependent feedback loop that facilitates the generation and accumulation of phosphatidylinositol 4,5-bisphosphate (PIP2) in microdomains on the PM. In turn, accumulating PIP2 may facilitate compartmentalization of Fwe to bulk endosomes, thereby terminating ADBE. I have a number of concerns.1) To visualize PIP2 at fly NMJs, the entire study relies only on a single PIP2 marker, the Gal4-driven expression of PLCδ1-PH-EGFP. Unfortunately, this marker not only reduces "free" PIP2 levels but also has dominant-negative effects on synaptic growth and SV cycling (Khuong et al., 2010). To properly support the notion that intense stimulation causes accumulation of PIP2 in microdomains on the PM, a number of corrections are required:a) the PLCδ1-PH-EGFP imaging experiments require a proper control, like the expression of PLCδ1-PHS39R-GFP, which does not bind PIP2. Measuring mCD-EGFP expression levels is not a proper control.

As suggested, we have now examined the expression and distribution of PLC_δ1_-PHS39R-EGFP at rest and after high K^+^ stimulation. We found that, unlike PLC_δ1_-PH-EGFP, PLC_δ1_-PHS39R-EGFP mainly localized in the cytosol of cells at rest. After intense stimulation, immunostaining signals of PLC_δ1_-PHS39R-EGFP did not accumulate on the plasma membrane (Figure 1—figure supplement 2C-D) and were not increased within the boutons (Figure 1—figure supplement 2E). Thus, the clustering of PLC_δ1_-PH-EGFP on the plasma membrane upon stimulation results from an increase in local PI(4,5)P_2_ concentration. These new data have been added to Figure 1—figure supplement 2C-E, and a respective statement has been added to the revised manuscript.

b) the stimulation-induced accumulation of PIP2 should be verified by a second independent marker like quantitative immunostainings for edogenous α-adaptin or other PIP2-binding proteins. This is especially important to rule out unaccounted dominant-negative effects.

As requested, we have now assessed recruitment of endogenous α-adaptin upon high K^+^ stimulation. We found that, similar to PLC_δ1_-PH-EGFP, levels of the α subunit of the AP-2 complex (AP-2α) were also increased on the plasma membrane upon high K^+^ stimulation (Figure 1—figure supplement 3A-B). The new data have been added to Figure 1—figure supplement 3A-B and a respective statement has been added to the revised manuscript.

c) It is unclear how changes in PLCδ1-PH-EGFP fluorescence (Figure 1A-B, D, plus all subsequent experiments using PLCδ1-PH-EGFP including Figure 3C-F, 4F-G) are measured at larval NMJs since no information about respective ROIs is provided. To make a strong statement about a PIP2 accumulation on the PM, PLCδ1-PH-EGFP fluorescence measured for an entire bouton (from one or a few slices) should be subtracted by the PLCδ1-PH-EGFP fluorescence obtained from the cytosol of the bouton. Measurements using ROIs for the entire bouton do not allow specific conclusions about PM levels.

We thank the reviewer for raising our oversight. As also suggested by reviewer #2, we have now re-quantified the signals of PLC_δ1_-PH-EGFP on the presynaptic plasma membranes, which were defined by α-Hrp staining.

For details of our revision, please see our response to point 4 in the comments of reviewer #2.

2) To significantly strengthen the conclusion that PIP2 accumulation drives ADBE initiation and SV reformation (Figure 2) a proper control should be used for the postulated PLCδ1-PH-EGFP-mediating reduction of PIP2 levels. Ideally, one may use expression of the PLCδ1-PHS39R-GFP mutant. Overexpression of synaptojanin alone does not make the point that the reduced ADBE levels are due to PIP2 depletion since synaptojanin also affects several other phosphatidylinositides.

We appreciate this suggestion. We conducted the recommended experiments and found that when larvae expressing PLC_δ1_-PHS39R-EGFP were grown at 29 ºC, the mutant protein failed to suppress ADBE (Figure 2C-D). The new data have been added to Figure 2C-D, and a respective statement has been added to the revised manuscript.

3) The proximity ligation experiments of Figure 3A require a proper control like PLCδ1-PHS39R-GFP co-expression with Flag-Fwe-HA. Otherwise, the observed proximity could be due to other factors.

As suggested, we have now performed a PLA assay to assess the association of Fwe with PLC_δ1_-PHS39R-EGFP. As shown in Figure 1—figure supplement 1 and 2, whereas PLC_δ1_-PH-EGFP mainly localized on the plasma membrane, PLC_δ1_-PHS39R-EGFP localized in the cytosol, both at rest and even upon high K^+^ stimulation. We further found that PLC_δ1_-PHS39R-EGFP indeed locates to the SVs marked by the SV proteins, Csp and Fwe (see Author response image 1). Consequently, when we did the PLA assay on the Flag-Fwe-HA rescue boutons expressing PLC_δ1_-PHS39R-EGFP, the close proximity between Fwe and SV-localized PLC_δ1_-PHS39R-EGFP due to 30-50 nm-sized SV already generated very strong PLA signals when the boutons were at rest and even after high K^+^ stimulation (see Author response image 2). Therefore, since PLC_δ1_-PH-EGFP and PLC_δ1_-PHS39R-EGFP have totally different subcellular localizations, we are not comfortable using PLC_δ1_-PHS39R-EGFP as a control protein in our PLA assay.

**Author response image 1. sa2fig1:** PLC_δ1_-PHS39R-EGFP locates to SVs marked by Csp or Fwe.

**Author response image 2. sa2fig2:** A close association between PLC_δ1_-PHS39R-EGFP and Flag-Fwe-HA on the SVs generates PLA signals at rest or after high K^+^ stimulation.

4) The images of Figure 3C, E show little evidence of PLCδ1-PH-EGFP microdomains at the PM, much in contrast to Figure 1. Does this mean that the quantification associated with these figures mostly reflects PIP2 on cytosolic bulk endosomes?

We apologize for the confusion. As also suggested by reviewer #2, we have replaced Z-projected images with single-plane images that focus on the central section of the boutons to represent protein distribution. PLC_δ1_-PH-EGFP microdomains were found mostly on the plasma membrane.

5) The shown knockdown data of CanA1 in Figure 3E-H require a control for specificity, best co-expression of an RNAi-resistant WT transgene.

The *UAS-canA1-RNAi (FB5)* used in our work was generated by cloning two copies of a region of CanA1 (320–1160 bp) in reverse orientation spaced by an 800 bp intron from CG7464 (Dijkers and O’Farrell, et al., 2007). It is therefore difficult to predict the functional small RNA fragments generated by this construct and thus to design an RNAi-resistant *canA1* transgene. As an alternative, we expressed another *UAS-canA1 RNAi* (JF01871) (Bloomington *Drosophila* Stock Center, BDSC#25850), which was generated by using different regions of the CanA1 coding region, the funcionality of which had been tested previously (Wong et al., 2014). We found that expression of *UAS-canA1 RNAi* (JF01871) using *nsyb-GAL4* suppressed PI(4,5)P_2_ microdomain formation (Figure 3H) and ADBE initiation (Figure 5A-B). These new data strengthen the role for CanA1 in PI(4,5)P_2_ and ADBE induction. These new data have been added in the figures indicated above, and respective statements have been added to the revised manuscript.

6) The study suggests that PIP2 binding to freshly exocytosed Fwe gates Fwe's Ca^2+^ channel activity. In turn, the resulting Ca^2+^ influx activates calcineurin and the postulated feedback loop accumulating PIP2 at the PM. If correct, one would expect that a mutation that several affects Fwe' Ca^2+^ channel activity (like FweE79Q) should significantly reduce the accumulation of PIP2. Is this indeed the case? If not, the Fwe-dependent PIP2 accumulation may occur through a different mechanism.

As suggested, we have now conducted experiments to determine if the Ca^2+^ channel activity of Fwe is involved in this process. We undertook rescue experiments based on our previous report in which *fwe* mutant boutons exhibited expression of the wild-type Fwe transgene or the FweE79Q mutant transgene that exhibits reduced Ca^2+^ conductance (Yao et al., 2017). We found that the wild-type Fwe transgene promoted PI(4,5)P_2_ microdomain formation, whereas the FweE79Q mutant transgene lost that ability (Figure 3E-F). Thus, Fwe triggers the formation of PI(4,5)P_2_ microdomains in a Ca^2+^-dependent manner. The new data have been added to Figure 3E-F, the detailed genotypes have been added to the respective figure legend, and an explanatory statement has been added to the revised manuscript.

7) The HA-Fwe-APEX2 experiments of Figure 6 require further controls including documentation that the fusion protein is properly localized to SVs and not excessively retained on the PM during ADBE. The EM alone is not convincing.

As suggested, we have now performed confocal imaging analysis to verify the SV localization of HA-Fwe-APEX2. As shown in Figure 6—figure supplement 1A, HA-Fwe-APEX2 (stained with α-Fwe) was highly colocalized with the specific SV markers, Syt and Csp. Furthermore, expression of this fusion protein was able to rescue the endocytic defects (Figures 4 and 5) and early animal lethality caused by loss of *fwe*. Therefore, HA-Fwe-APEX2 is functionally equivalent to endogenous Fwe. In our TEM images (Figure 6—figure supplement 1C-D), DAB signals generated by HA-Fwe-APEX2 were found on SVs (yellow arrows) and on the plasma membrane (white arrows) under both resting and stimulation conditions. These new data have been added to Figure 6—figure supplement 1, and a respective statement has been added to the revised manuscript.

8) The co-immunoprecipitation of Fwe with AP2 shown in Figure 6G requires quantification.

As suggested by reviewer #1, since the data on Fwe and AP-2 complex association (previous Figure 6G-H and text) was deemed superfluous, we have now removed this data from the revised manuscript.

9) The evidence for the conclusion that accumulating PIP2 facilitates "sorting" of Fwe for ADBE is weak because it essentially rests on the HA-Fwe-APEX2 experiment. Since reducing PIP2 levels by synaptojanin expression reduced the bulk endosome localization of Fwe as judged by DAB staining levels (Figure 6D-F), the authors conclude that "PIP2 microdomains can facilitate the retrieval of Fwe to the bulk endosome". However, this appears to be somewhat a catch22 since reduced PIP2 levels decrease the overall amount of bulk endocytosis. Whether the potentially reduced levels of internalized Fwe are disproportionally reduced has not been addressed.

We thank the reviewer for raising this point. In our data (Figure 6C-E), we controlled animal growth temperature at 25 °C to express a mild level of Synj. Under this condition, the suppressive effect on the formation of PI(4,5)P_2_ microdomains (Figure 2B) is much stronger than that on ADBE initiation (Figure 2C-D), and increasing Synj expression further abolished ADBE (Figure 2C-D). Therefore, our data argue that reduced Fwe localization in the bulk endosome is unlikely to be due to a reduction in the overall amount of bulk endocytosis. Instead, PI(4,5)P_2_ plays a role in retrieval of Fwe to the bulk endosomes.

In addition, DAB staining intensity should not be used to quantify protein levels as the enzymatic reaction may have different constrains on SVs versus endosomes.

Upon H2O2 activation, APEX2 catalyzes DAB polymerization, and DAB polymers proximally label the phospholipids where the APEX2-fused proteins are closely localized. Synaptic vesicles and bulk endosomes are generated freshly from the plasma membrane upon exocytosis. Therefore, both vesicular compartments are expected to hold very similar compositions of phospholipids, glycolipids, and cholesterols, even though they also contain distinct types of phosphoinositides as minor components. We certainly agree that the enzymatic reaction may have different constraints on SVs versus endosomes. It has not been demonstrated that the lipid and protein composition of the membrane affects DAB labeling. Therefore, we assume that the DAB staining intensity is positively related to the membrane density of the APEX2-fused Fwe protein on both bulk endosomes and SVs. Even so, to ease the reviewer’s concern, we have now re-quantified the DAB signal intensity ratio of bulk endosome to SV to indicate the protein levels of HA-Fwe-APEX2 on bulk endosomes, relative to those on the SVs. With this normalization, we can verify the role of PI(4,5)P_2_ microdomains in Fwe localization on bulk endosomes. We provide details of description and data quantification in the revised main text, Materials and methods, and figure legends as follows:

“To minimize staining variability from bouton to bouton, we compared DAB staining intensities on bulk endosomes and SVs from the same boutons. As shown in Figure 6B-F, compared to Flag-Fwe-HA rescue boutons, in HA-Fwe-APEX2 rescue boutons, the staining intensities of bulk endosomes were more abundant compared to those of the surrounding SVs, revealing a mechanism by which Fwe is recycled to bulk endosomes after it initiates ADBE.”

When we perturbed PI(4,5)P_2_ microdomains by Synj overexpression (Figure 2B), the bulk endosome localization of HA-Fwe-APEX2 was significantly reduced (Figure 6D-F), but there was only a mild reduction in SV localization (Figure 6D-E). Therefore, in addition to initiating ADBE, PI(4,5)P_2_ microdomains play a role in facilitating the retrieval of Fwe to the bulk endosome, enabling ADBE to remove its trigger via a negative feedback regulatory mechanism, thereby reducing endocytosis to prevent excess membrane uptake. We now provide this explanation in the revised manuscript.

For data quantification of DAB intensities, the display color of TEM images was reverted to grayscale using ImageJ. Average DAB staining intensity for each individual bulk endosome was assessed. Then the average DAB staining intensities of 50-100 surrounding SVs from the same bouton image were used to assess the relative level of HA-Fwe-APEX2 on bulk endosomes vs SVs.

We add quantification data for the DAB staining intensity ratio of bulk endosomes to SVs the surrounding SVs in Figure 6F. The number of bulk endosomes, NMJ boutons, and larvae counted (Flag-Fwe-HA rescue control: Bulk endosomes (n=33) derived from 5 NMJ boutons of two different larvae; HA-Fwe-APEX2 rescue: Bulk endosomes (n=47) derived from 6 NMJ boutons of two different larvae; HA-Fwe-APEX2 rescue expressing Synj: Bulk endosomes (n=59) derived from 9 NMJ boutons of two different larvae. Individual values were shown in graphs. This description is now stated in figure legends.

[Editors' note: further revisions were suggested prior to acceptance, as described below.]

The authors have provided a meaningfully improved manuscript that largely addresses the major concerns of the paper. In particular, the authors provide several new lines of evidence that levels or organization of PI(4,5)P2 are likely changing in response to stimulation: There is no change in the new negative control PH S39 mutant, AP2a levels increase at the membrane, and the levels of native and anti-GFP fluorescence of PH-PLC increase at the membrane. Especially given the strong correspondence between in vitro and in vivo data and the consistent phenotypes among the perturbations in the proposed pathway, these new data further strengthen the main claim that Fwe acts in a positive feedback loop with Ca++ and PI(4,5)P2.It is worth noting that the precise nature of the microdomains remains a bit unclear. It is difficult to discern whether PI(4,5)P2 microdomains are forming de novo following stimulation, or whether PI(4,5)P2 ¬signal increases in existing microdomains (for example, AP2a recruitment to the membrane is quite high even under resting conditions). And as noted by the authors, while the PH domain is the best available tool to answer the questions posed, it remains possible that properties of the microdomains partially reflect the function-perturbing aspects of the probe. However, these caveats do not significantly detract from the core conclusions of the paper.Figure 3H. Something is wrong with the shown KD data by UAS-canA1-RNAi(FB5) or the drawn conclusion that "reducing CanA1 levels via expression of either RNAi construct greatly suppressed the formation of PI(4,5)P2 microdomains compared to the control (Figure 3G-H)". The resting levels for the canA1 KD are only 50% of control. The K^+^-induced increase PLCδ1-PH-EGFP in the KD is as large as in control. Accordingly, the KD seems to have no effect, which would suggest that one of the two KDs is not specific for canA1.If the FB5 KD is not working, the one needs a wild type CanA1 co-expression for the KD with the canA1-RNAi(JF01871) transgene to demonstrate specificity.Please clarify.Figure 3I. The canA1 KD uses the controversial UAS-canA1-RNAi(FB5) transgene, which does not appear to block the formation of PI(4,5)P2 microdomains (Figure 3H). The experiment should be done with the UAS-canA1-RNAi(JF01871) transgene as only this transgene reliable affects PI(4,5)P2 microdomains (assuming the KD can be rescued). Otherwise, one cannot conclude that "Calcineurin mediates the Ca^2+^ influx conducted by Fwe to induce PI(4,5)P2 microdomains". Resolving this issue is critical.

For suggested rescue experiment, we have carefully gone through public resource indicated in Flybase. Unfortunately, a wild-type *canA1* transgene is not available. If we would begin the experiment by establishing a wild-type canA1 transgene, it will take at least 3-4 months to be able to get the result. Alternatively, we have used a *UAS* transgene of a constitutively active form of CanA1 (UAS-CanA1.δ.HA(4B)), which was published by Dijkers and O’Farrell, 2007, and is available in Bloomington *Drosophila* Stock Center (#83343). However, we observed that expression of UAS-CanA1.δ.HA(4B) alone already inhibited the formation of PI(4,5)P_2_ microdomains in high K^+^ stimulation and expression of PI(4,5)P_2_ at rest (see Author response image 3). Therefore, sustained high CanA1 activity can also perturb the PI(4,5)P_2_ increase after strong stimulation and even lower the resting PI(4,5)P_2_ concentrations. One would expect that expression of a wild-type canA1 transgene likely causes similar effects. Therefore, we are not comfortable to assess the rescue effect by constitutively increasing CanA1 enzymatic activity if dynamic CanA1 activity is needed to promote PI(4,5)P_2_ microdomain formation.

**Author response image 3. sa2fig3:** 

To address the reviewer’s concern, we have tested another independent line of UAS-canA1-RNAi (FB)(Dijkers and O’Farrell, 2007), UAS-canA1-RNAi (FB4). FB4 and FB5 lines carries identical UAS-canA1-RNAi transgene, but their canA1-RNAi transgenes are inserted on different chromosomes, e.g. second chromosome for FB4 and third chromosome for FB5. Unlike UAS-canA1-RNAi (FB5), we found that expression of UAS-canA1-RNAi (FB4) significantly impaired the formation of PI(4,5)P_2_ microdomains but did not affect the resting level of PI(4,5)P_2_ (Figure 3H), which is close to the effects of UAS-canA1-RNAi (JF). Thus, the FB5 line may have an unclean genetic background which influences the resting levels of PI(4,5)P_2_. In addition, we have examined the effect of expression of both UAS-CanA1-RNAi (JF01871) and UAS-CanA1-RNAi (FB4) on intracellular Ca^2+^ increase upon high K^+^ stimulation. As shown in Figure 3I-J, expression of UAS-CanA1-RNAi (JF01871) or UAS-CanA1-RNAi (FB4) was able to suppress intracellular Ca^2+^ increase upon high K^+^ stimulation. We hope that the new data of two independent RNAi lines we added can support the role of CanA1 in the PI(4,5)P_2_ microdomain formation. We have done the best we can do during limited revision time. The new data have been added to Figure 3H-J, and a respective statement has been added to the revised manuscript. Accordingly, we have removed the results of UAS-canA1-RNAi (FB5) from Figures 3 and 5.

"Together, these data suggest that the PI(4,5)P2 microdomains initiate ADBE to control SV membrane retrieval upon intense stimulation." This sentence should be rephrased. The shown FM1-43 data (Figure 2—figure supplement 2B) only suggest that overexpression of PLCδ1-PH-EGFP or Synj impairs SV endocytosis independent of the mode of endocytosis. The shown FM1-43 data do not exclude a role of PI(4,5)P2 for other modes of endocytosis.Moreover, the FM1-43 data use a much shorter (30 seconds) and milder stimulation than the solid EM data visualizing ADBE (10 minutes; Figure 2). Therefore, the FM1-43 and EM data are not directly comparable.The same concern applies to the FM1-43-visulaized SV exocytosis data, which indicate that SV exocytosis is not impaired by overexpression of PLCδ1-PH-EGFP or Synj if endocytosis is stimulated for 1 minute. Again, the data are not comparable to the ADBE defect shown in Figure 2. I suggest either to omit the data of Figure 2—figure supplement 2 or use the same stimulation paradigm for the FM1-43 and EM data.

As suggested, we have removed Figure 2—figure supplement 2 and the statement.